Corrected: Publisher correction

# Ancient DNA from Chalcolithic Israel reveals the role of population mixture in cultural transformation

Éadaoin Harney[1,2,3], Hila May[4,5], Dina Shalem[6], Nadin Rohland[2], Swapan Mallick[2,7,8], Iosif Lazaridis[2,3], Rachel Sarig[5,9], Kristin Stewardson[2,8], Susanne Nordenfelt[2,8], Nick Patterson[7,8], Israel Hershkovitz[4,5] & David Reich[2,3,7,8]

The material culture of the Late Chalcolithic period in the southern Levant (4500–3900/3800 BCE) is qualitatively distinct from previous and subsequent periods. Here, to test the hypothesis that the advent and decline of this culture was influenced by movements of people, we generated genome-wide ancient DNA from 22 individuals from Peqi'in Cave, Israel. These individuals were part of a homogeneous population that can be modeled as deriving ~57% of its ancestry from groups related to those of the local Levant Neolithic, ~17% from groups related to those of the Iran Chalcolithic, and ~26% from groups related to those of the Anatolian Neolithic. The Peqi'in population also appears to have contributed differently to later Bronze Age groups, one of which we show cannot plausibly have descended from the same population as that of Peqi'in Cave. These results provide an example of how population movements propelled cultural changes in the deep past.

[1] Department of Organismic and Evolutionary Biology, Harvard University, Cambridge, MA 02138, USA. [2] Department of Genetics, Harvard Medical School, Boston, MA 02115, USA. [3] The Max Planck–Harvard Research Center for the Archaeoscience of the Ancient Mediterranean, Cambridge, MA 02138, USA. [4] Department of Anatomy and Anthropology, Sackler Faculty of Medicine, Tel Aviv University, Tel Aviv 6997801, Israel. [5] Shmunis Family Anthropology Institute, Dan David Center for Human Evolution and Biohistory Research, Sackler Faculty of Medicine, Steinhardt Natural History Museum, Tel Aviv University, Tel Aviv 6997801, Israel. [6] The Institute for Galilean Archaeology, Kinneret Academic College, Kinneret 15132, Israel. [7] Broad Institute of MIT and Harvard, Cambridge 02142 MA, USA. [8] Howard Hughes Medical Institute, Boston, MA 02115, USA. [9] The Maurice and Gabriela Goldschleger School of Dental Medicine, Sackler Faculty of Medicine, Tel Aviv University, Tel Aviv 6997801, Israel. These authors contributed equally: Éadaoin Harney, Hila May. These authors jointly supervised this work: Israel Hershkovitz, David Reich. Correspondence and requests for materials should be addressed to É.H. (email: eadaoinharney@gmail.com) or to H.M. (email: mayhila@post.tau.ac.il)

The material culture of the Late Chalcolithic period in the southern Levant contrasts qualitatively with that of earlier and later periods in the same region. The Late Chalcolithic in the Levant is characterized by increases in the density of settlements, introduction of sanctuaries[1–3], utilization of ossuaries in secondary burials[4,5], and expansion of public ritual practices as well as an efflorescence of symbolic motifs sculpted and painted on artifacts made of pottery, basalt, copper, and ivory[6–9]. The period's impressive metal artifacts, which reflect the first known use of the "lost wax" technique for casting of copper, attest to the extraordinary technical skill of the people of this period[10,11].

The distinctive cultural characteristics of the Late Chalcolithic period in the Levant (often related to the Ghassulian culture, although this term is not in practice applied to the Galilee region where this study is based) have few stylistic links to the earlier or later material cultures of the region, which has led to extensive debate about the origins of the people who made this material culture. One hypothesis is that the Chalcolithic culture in the region was spread in part by immigrants from the north (i.e., northern Mesopotamia), based on similarities in artistic designs[12,13]. Others have suggested that the local populations of the Levant were entirely responsible for developing this culture, and that any similarities to material cultures to the north are due to borrowing of ideas and not to movements of people[2,14–19].

To explore these questions, we studied ancient DNA from a Chalcolithic site in Northern Israel, Peqi'in (Fig. 1a). This cave, which is around 17 m long and 4.5–8.0 m wide (Fig. 1b), was discovered during road construction in 1995, and was sealed by natural processes during or around the end of the Late Chalcolithic period (around 3900 BCE). Archeological excavations have revealed an extraordinary array of finely crafted objects, including chalices, bowls, and churns, as well as more than 200 ossuaries and domestic jars repurposed as ossuaries (the largest number ever found in a single cave), often decorated with anthropomorphic designs (Fig. 1c)[20,21]. It has been estimated that the burial cave contained up to 600 individuals[22], making it the largest burial site ever identified from the Late Chalcolithic period in the Levant. Direct radiocarbon dating suggests that the cave was in use throughout the Late Chalcolithic (4500–3900 BCE), functioning as a central burial location for the region[21,23].

Previous genome-wide ancient DNA studies from the Near East have revealed that at the time when agriculture developed, populations from Anatolia, Iran, and the Levant were approximately as genetically differentiated from each other as present-day Europeans and East Asians are today[24,25]. By the Bronze Age, however, expansion of different Near Eastern agriculturalist populations—Anatolian, Iranian, and Levantine—in all directions and admixture with each other substantially homogenized populations across the region, thereby contributing to the relatively low genetic differentiation that prevails today[24]. Lazaridis et al.[24] showed that the Levant Bronze Age population from the site of 'Ain Ghazal, Jordan (2490–2300 BCE) could be fit statistically as a mixture of around 56% ancestry from a group related to Levantine Pre-Pottery Neolithic agriculturalists (represented by ancient DNA from Motza, Israel and 'Ain Ghazal, Jordan; 8300–6700 BCE) and 44% related to populations of the Iranian Chalcolithic (Seh Gabi, Iran; 4680–3662 calBCE). Haber et al.[26] suggested that the Canaanite Levant Bronze Age population from the site of Sidon, Lebanon (~1700 BCE) could be modeled as a mixture of the same two groups albeit in different proportions (48% Levant Neolithic-related and 52% Iran Chalcolithic-related). However, the Neolithic and Bronze Age sites analyzed so far in the Levant are separated in time by more than three thousand years, making the study of samples that fill in this gap, such as those from Peqi'in, of critical importance.

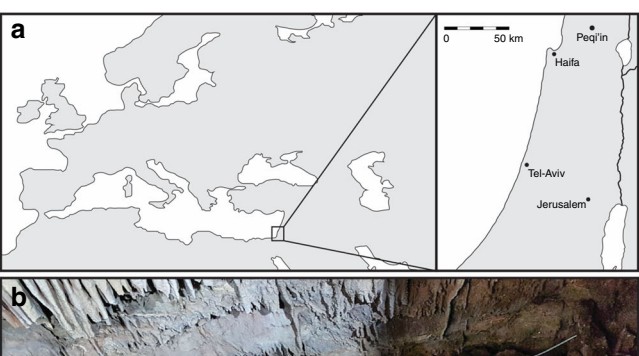

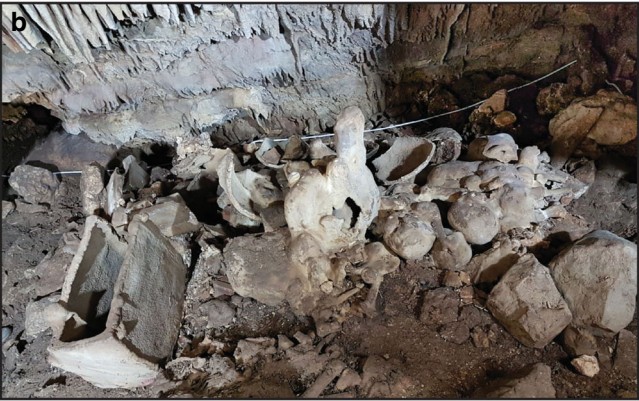

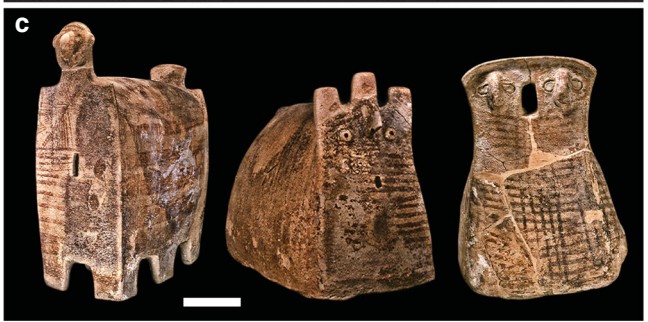

**Fig. 1** Site background. **a** Location of the site of Peqi'in Cave. **b** Photo of the interior of Peqi'in Cave. Photo Hila May, courtesy of the Dan David Center of Human Evolution and Biohistory. **c** Photo of several burial urns (ossuaries) from Peqi'in Cave. Scale bar: 10 cm. Photo Mariana Salzberger, courtesy of the Israel Antiquities Authority. 

In a dedicated clean room facility at Harvard Medical School, we obtained bone powder from 48 skeletal remains, of which 37 were petrous bones known for excellent DNA preservation[27]. We extracted DNA[28] and built next-generation sequencing libraries to which we attached unique barcodes to minimize the possibility of contamination. We treated the libraries with Uracil–DNA glycosylase (UDG) to reduce characteristic ancient DNA damage at all but the first and last nucleotides[29] (Supplementary Table 1 and Supplementary Data 1 provide background for successful samples and report information for each library, respectively). After initial screening by enriching the libraries for mitochondrial DNA, we enriched promising libraries for sequences overlapping about 1.2 million single nucleotide polymorphisms (SNPs)[30,31]. We evaluated each individual for evidence of authentic ancient DNA by limiting to libraries with a minimum of 3% cytosine-to-thymine errors at the final nucleotide[29], by requiring that the ratio of X-to-Y-chromosome sequences was characteristic of either a male or a female, by requiring >95% matching to the consensus sequence of mitochondrial DNA[30], and by requiring (for males) a lack of variation at known polymorphic positions on chromosome X (point estimates of contamination of less than 2%)[32]. We also restricted to individuals with at least 5000 of the targeted SNPs covered at least once.

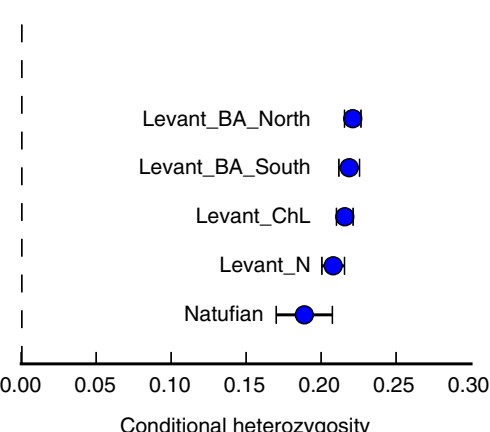

**Fig. 2** Genetic diversity in the ancient Levant. Heterozygosity increases in ancient Levantine populations over time. The estimated statistic ± 3 standard errors is indicated

This procedure produced genome-wide data from 22 ancient individuals from Peqi'in Cave (4500–3900 calBCE), with the individuals having a median of 358,313 of the targeted SNPs covered at least once (range: 25,171–1,002,682). The dataset is of exceptional quality given the typically poor preservation of DNA in the warm Near East, with a higher proportion of samples yielding appreciable coverage of ancient DNA than has previously been obtained from the region, likely reflecting the optimal sampling techniques we used and the favorable preservation conditions at the cave. We analyzed this dataset in conjunction with previously published datasets of ancient Near Eastern populations[24,26] to shed light on the history of the individuals buried in the Peqi'in cave site, and on the population dynamics of the Levant during the Late Chalcolithic period.

## Results

**Genetic differentiation and diversity in the ancient Levant**. A total of 20 Peqi'in samples appear to be unrelated to each other to the limits of our resolution (that is, genetic analysis suggested that they were not first, second, or third degree relatives of each other), and we used these as our analysis set. Taking advantage of the new data point added by the Peqi'in samples, we began by studying how genetic differentiation among Levantine populations changed over time. We replicate previous reports of dramatic decline in genetic differentiation over time in West Eurasia[24], observing a median pairwise $F_{ST}$ of 0.023 (range: 0.009–0.061) between the Peqi'in samples (abbreviation: Levant_ChL) and other West Eurasian Neolithic and Chalcolithic populations, relative to a previously reported median pairwise $F_{ST}$ of 0.098 (range: 0.023–0.153) observed between populations in pre-Neolithic periods, 0.015 (range: 0.002–0.045) in the Bronze Age periods, and 0.011 (range: 0–0.046) in present-day West Eurasian populations[24]. Thus, the collapse to present-day levels of differentiation was largely complete by the Chalcolithic (Supplementary Figure 1).

We also observe an increase in genetic diversity over time in the Levant as measured by the rate of polymorphism between two random genome sequences at each SNP analyzed in our study. Specifically, the Levant_ChL population exhibits an intermediate level of heterozygosity relative to the earlier and later populations (Fig. 2).

Both the increasing genetic diversity over time, and the reduced differentiation between populations as measured via $F_{ST}$,

are consistent with a model in which gene flow reduced differentiation across groups while increasing diversity within groups.

**Genetic affinities of the individuals of Peqi'in Cave**. To obtain a qualitative picture of how these individuals relate to previously published ancient DNA and to present-day people, we began by carrying out principal component analysis (PCA)[33]. In a plot of the first and second principal components (Fig. 3a), the samples from Peqi'in Cave form a tight cluster, supporting the grouping of these individuals into a single analysis population (while we use the broad name "Levant_ChL" to refer to these samples, we recognize that they are currently the only ancient DNA available from the Levant in this time period and future work will plausibly reveal genetic substructure in Chalcolithic samples over the broad region). The Levant_ChL cluster overlaps in the PCA with a cluster containing Neolithic Levantine samples (Levant_N), although it is slightly shifted upward on the plot toward a cluster corresponding to samples from the Levant Bronze Age, including samples from 'Ain Ghazal, Jordan (Levant_BA_South) and Sidon, Lebanon (Levant_BA_North). The placement of the Levant_ChL cluster is consistent with a previously observed pattern whereby chronologically later Levantine populations are shifted towards the Iran Chalcolithic (Iran_ChL) population compared to earlier Levantine populations, Levant_N (Pre-Pottery and Pottery Neolithic agriculturalists from present-day Israel and Jordan) and Natufians (Epipaleolithic hunter-gatherers from present-day Israel)[24].

ADMIXTURE model-based clustering analyses[34] produced results consistent with PCA in suggesting that individuals from the Levant_ChL population had a greater affinity on average to Iranian agriculturalist-related populations than was the case for earlier Levantine individuals. Figure 3b shows the ADMIXTURE results for the ancient individuals assuming $K = 11$ clusters (we selected this number because it maximizes ancestry components that are correlated to ancient populations from the Levant, from Iran, and European hunter-gatherers)[24]. Like all Levantine populations, the primary ancestry component assigned to the Levant_ChL population, shown in blue, is maximized in earlier Levant_N and Natufian individuals. ADMIXTURE also assigns a component of ancestry in Levant_ChL, shown in green, to a population that is generally absent in the earlier Levant_N and Natufian populations, but is present in later Levant_BA_South and Levant_BA_North samples. This green component is also inferred in small proportions in several samples assigned to the Levant_N, but there is not a clear association to archaeological location or date, and these individuals are not significantly genetically distinct from the other individuals included in Levant_N by formal testing, and thus we pool all Levant_N for the primary analyses in this study (Supplementary Note 1)[24].

**Population continuity and admixture in the Levant**. To determine the relationship of the Levant_ChL population to other ancient Near Eastern populations, we used $f$-statistics[35] (see Supplementary Note 2 for more details). We first evaluated whether the Levant_ChL population is consistent with descending directly from a population related to the earlier Levant_N. If this was the case, we would expect that the Levant_N population would be consistent with being more closely related to the Levant_ChL population than it is to any other population, and indeed we confirm this by observing positive statistics of the form $f_4$ (Levant_ChL, A; Levant_N, Chimpanzee) for all ancient test populations, $A$ (Fig. 4a). However, Levant_ChL and Levant_N population do not form a clade, as when we compute symmetry statistics of the form $f_4$ (Levant_N, Levant_ChL; A, Chimpanzee),

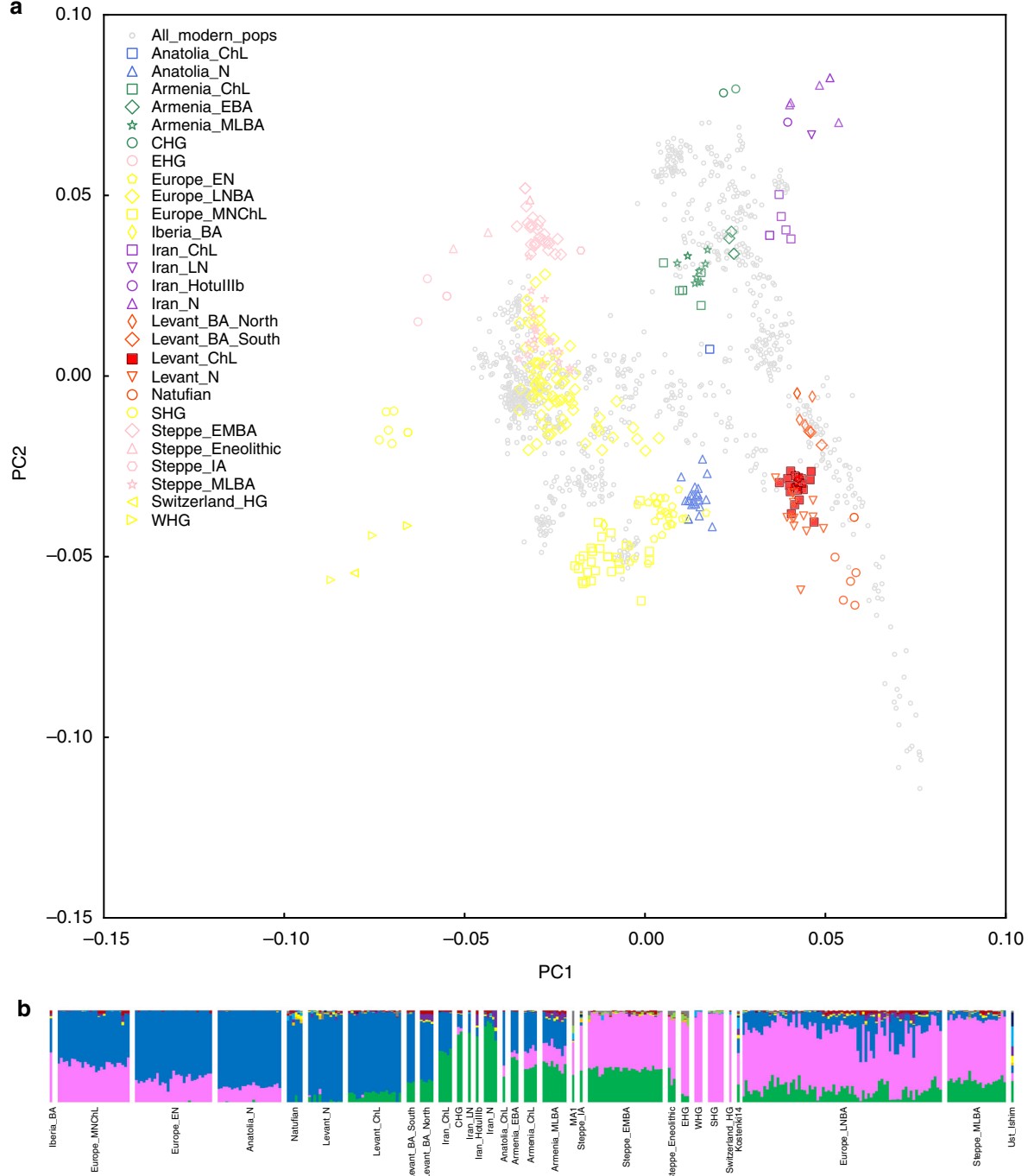

**Fig. 3** Genetic structure of analyzed individuals. **a** Principal component analysis of 984 present-day West Eurasians (shown in gray) with 306 ancient samples projected onto the first two principal component axes and labeled by culture. **b** ADMIXTURE analysis of 984 and 306 ancient samples with $K = 11$ ancestral components. Only ancient samples are shown

we find that the statistic is often negative, with Near Eastern populations outside the Levant sharing more alleles with Levant_ChL than with Levant_N (Fig. 4b). We conclude that while the Levant_N and Levant_ChL populations are clearly related, the Levant_ChL population cannot be modeled as descending directly from the Levant_N population without additional admixture related to ancient Iranian agriculturalists. Direct evidence that Levant_ChL is admixed comes from the statistic $f_3$ (Levant_ChL; Levant_N, $A$), which for some populations, $A$, is significantly negative indicating that allele frequencies in Levant_ChL tend to be intermediate between those in Levant_N and $A$—a pattern that can only arise if Levant_ChL is the product

of admixture between groups related, perhaps distantly, to Levant_N and $A$[35]. The most negative $f_3$- and $f_4$-statistics are produced when $A$ is a population from Iran or the Caucasus. This suggests that the Levant_ChL population is descended from a population related to Levant_N, but also harbors ancestry from non-Levantine populations related to those of Iran or the Caucasus that Levant_N does not share (or at least share to the same extent).

**The ancestry of the Levant Chalcolithic people**. We used qpAdm as our main tool for identifying plausible admixture

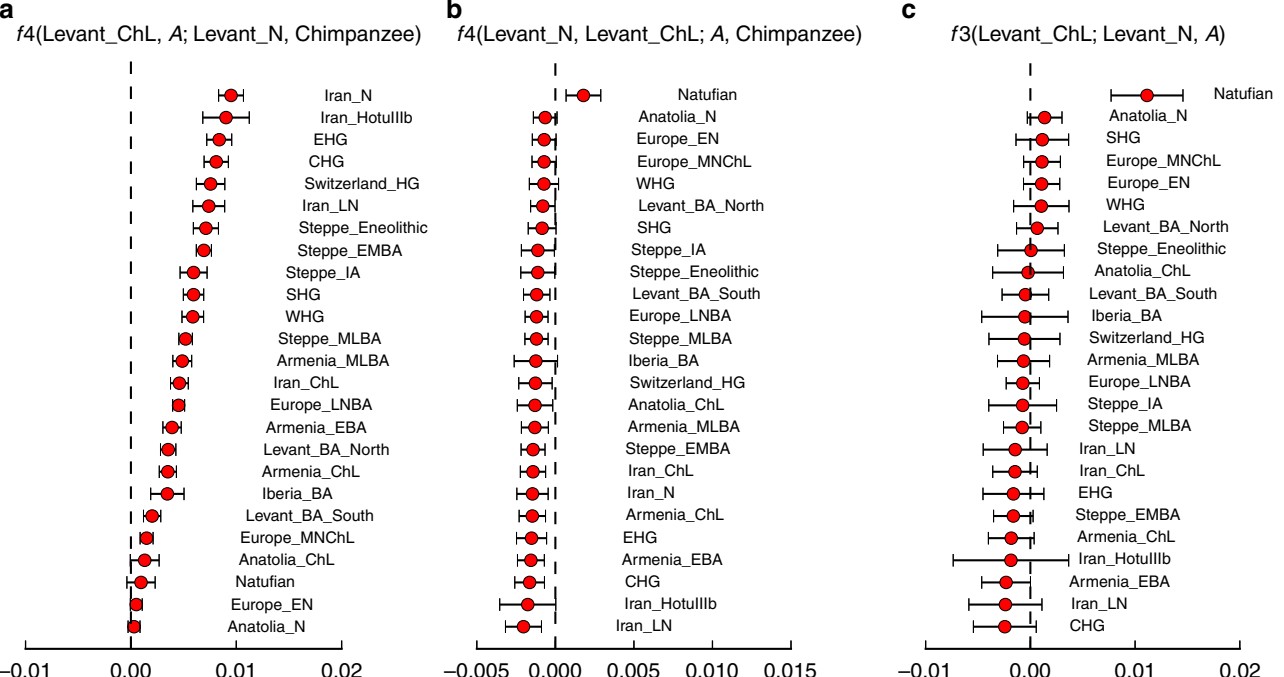

**Fig. 4** Genetic characteristics of the Levant_ChL. **a** The statistic $f_4$(Levant_ChL, A; Levant_N, Chimpanzee) demonstrates a close relationship between the Neolithic and Chalcolithic Levant populations, as the Levant Neolithic shares more alleles with the Levant Chalcolithic than with any other populations. **b** The statistic $f_4$(Levant_N, Levant_ChL; A, Chimpanzee) shows an asymmetrical relationship between Levant_N and Levant_ChL and other ancient West Eurasian populations. The statistic is most negative for populations from Iran and the Caucasus, indicating that Levant_ChL shares more alleles with them than does Levant_N. **c** The statistic $f_3$(Levant_ChL; Levant_N, A) tests for signals of admixture in Levant_ChL. Negative $f_3$-statistics indicate that the Levant_ChL population is admixed. Populations from Iran and the Caucasus produce the most negative statistics. The estimated statistic ± 3 standard errors is indicated

models for the ancient populations for which we have data (see Supplementary Note 3 for more details)[36].

The qpAdm method evaluates whether a tested set of N "Left" populations—including a "target" population (the population whose ancestry is being modeled) and a set of N − 1 additional populations—are consistent with being derived from mixtures in various proportions of N − 1 ancestral populations related differentially to a set of outgroup populations, referred to as "Right" populations. For all our analyses, we use a base set of 11 "Right" outgroups referred to collectively as "09NW"—Ust_Ishim, Kostenki14, MA1, Han, Papuan, Onge, Chukchi, Karitiana, Mbuti, Natufian, and WHG—whose value for disentangling divergent strains of ancestry present in ancient Near Easterners has been documented in Lazaridis et al.[24] (for some analyses we supplement this set with additional outgroups). To evaluate whether the "Left" populations are consistent with a hypothesis of being derived from N − 1 sources, qpAdm effectively computes all possible statistics of the form $f_4$(Left$_i$, Left$_j$; Right$_k$, Right$_l$), for all possible pairs of populations in the proposed "Left" and "Right" sets. It then determines whether all the statistics can be written as a linear combination of $f_4$-statistics corresponding to the differentiation patterns between the proposed N − 1 ancestral populations, appropriately accounting for the covariance of these statistics and computing a single p value for fit based on a Hotelling T-squared distribution[36]. For models that are consistent with the data (p > 0.05), qpAdm estimates proportions of admixture for the target population from sources related to the N − 1 ancestral populations (with standard errors). Crucially, qpAdm does not require specifying an explicit model for how the "Right" outgroup populations are related.

We first examined all possible "Left" population sets that consisted of Levant_ChL along with one other ancient population

from the analysis dataset. Testing a wide range of ancient populations, we found that p values for all possible Left populations were below 0.05 (Supplementary Data 2), showing that Levant_ChL is not consistent with being a clade with any of them relative to the "Right" 09NW outgroups. We then considered models with "Left" population sets containing Levant_ChL along with two additional ancient populations, which corresponds to modeling the Levant_ChL as the result of a two-way admixture between populations related to these two other ancient populations. To reduce the number of hypotheses tested, we restricted the models to pairs of source populations that contain at least one of the six populations that we consider to be the most likely admixture sources based on geographical and temporal proximity: Anatolia_N, Anatolia_ChL, Armenia_ChL, Iran_ChL, Iran_N, and Levant_N. Again, we find no plausible two-way admixture models using a p > 0.05 threshold (Supplementary Figure 2 and Supplementary Data 3). Finally, we tested possible three-way admixture events, restricting to triplets that contain at least two of the six most likely admixture sources. Plausible solutions at p > 0.05 are listed in Table 1 (full results are reported in Supplementary Figure 3 and Supplementary Data 4).

We found multiple candidates for three-way admixture models, always including (1) Levant_N (2) either Anatolia_N or Europe_EN and (3) either Iran_ChL, Iran_N, Iran_LN, Iran_-HotuIIIb or Levant_BA_North. These are all very similar models, as Europe_EN (early European agriculturalists) are known to be genetically primarily derived from Anatolian agriculturalists (Anatolia_N)[31], and Levant_BA_North has ancestry related to Levant_N and Iran_ChL[26]. To distinguish between models involving Anatolian Neolithic (Anatolia_N) and European Early Neolithic (Europe_EN), we repeated the analysis including additional outgroup populations in the "Right" set that are

**Table 1 Plausible models of Levant_ChL as a mixture of three sources**

| Target | Source left populations | | | Outgroup right pops | p Value rank = 2 | Admixture proportions | | | Standard error | | |
|---|---|---|---|---|---|---|---|---|---|---|---|
| | A | B | C | | | A | B | C | A | B | C |
| *Levant_ChL* | *Levant_N* | *Anatolia_N* | *Iran_ChL* | *09NW* | *6.82E−02* | *0.580* | *0.259* | *0.161* | *0.035* | *0.048* | *0.030* |
| *Levant_ChL* | *Levant_N* | *Anatolia_N* | *Iran_ChL* | *09NWL* | *6.69E−02* | *0.561* | *0.291* | *0.148* | *0.030* | *0.037* | *0.027* |
| *Levant_ChL* | *Levant_N* | *Anatolia_N* | *Iran_ChL* | *09NWLY* | *9.15E−02* | *0.561* | *0.301* | *0.138* | *0.030* | *0.032* | *0.020* |
| **Levant_ChL** | **Levant_N** | **Anatolia_N** | **Iran_ChL** | **09NWSGEITMLY** | **1.14E−01** | **0.571** | **0.264** | **0.166** | **0.030** | **0.028** | **0.015** |
| *Levant_ChL* | *Levant_N* | *Anatolia_N* | *Iran_LN* | *09NW* | *1.14E−01* | *0.562* | *0.281* | *0.157* | *0.035* | *0.044* | *0.028* |
| Levant_ChL | Levant_N | Anatolia_N | Iran_LN | 09NWL | 4.62E−02 | 0.534 | 0.335 | 0.131 | 0.030 | 0.032 | 0.025 |
| Levant_ChL | Levant_N | Anatolia_N | *Iran_HotuIIIb* | *09NW* | *8.22E−02* | *0.484* | *0.217* | *0.299* | *0.033* | *0.043* | *0.045* |
| Levant_ChL | Levant_N | Anatolia_N | Iran_HotuIIIb | 09NWL | 6.11E−03 | 0.476 | 0.305 | 0.219 | 0.036 | 0.035 | 0.048 |
| *Levant_ChL* | *Levant_N* | *Anatolia_N* | *Iran_N* | *09NW* | *1.09E−01* | *0.579* | *0.308* | *0.113* | *0.035* | *0.041* | *0.020* |
| Levant_ChL | Levant_N | Anatolia_N | Iran_N | 09NWL | 4.74E−02 | 0.551 | 0.356 | 0.094 | 0.030 | 0.031 | 0.018 |
| *Levant_ChL* | *Levant_N* | *Anatolia_N* | *Levant_BA_North* | *09NW* | *2.16E−01* | *0.494* | *0.233* | *0.273* | *0.035* | *0.047* | *0.046* |
| *Levant_ChL* | *Levant_N* | *Anatolia_N* | *Levant_BA_North* | *09NWL* | *1.13E−01* | *0.478* | *0.286* | *0.236* | *0.034* | *0.036* | *0.478* |
| Levant_ChL | Levant_N | Anatolia_N | Levant_BA_North | 09NWLY | 3.36E−02 | 0.481 | 0.254 | 0.265 | 0.034 | 0.034 | 0.040 |
| *Levant_ChL* | *Levant_N* | *Europe_EN* | *Iran_ChL* | *09NW* | *6.80E−02* | *0.633* | *0.172* | *0.195* | *0.029* | *0.033* | *0.027* |
| Levant_ChL | Levant_N | Europe_EN | Iran_ChL | 09NWL | 6.38E−03 | 0.613 | 0.207 | 0.180 | 0.028 | 0.030 | 0.027 |
| *Levant_ChL* | *Levant_N* | *Europe_EN* | *Iran_N* | *09NW* | *6.89E−02* | *0.647* | *0.213* | *0.140* | *0.029* | *0.031* | *0.020* |
| Levant_ChL | Levant_N | Europe_EN | Iran_N | 09NWL | 1.88E−03 | 0.628 | 0.262 | 0.109 | 0.029 | 0.027 | 0.200 |

Note: Populations that produce p values greater than 0.05 with plausible admixture proportions (0–1) are highlighted in italics. The model with the lowest standard errors is indicated in bold

sensitive to the European hunter-gatherer-related admixture present to a greater extent in Europe_EN than in Anatolia_N (Supplementary Figure 4a)[31] (thus, we added Switzerland_HG, SHG, EHG, Iberia_BA, Steppe_Eneolithic, Europe_MNChL, Europe_LNBA to the "Right" outgroups; abbreviations in Supplementary Table 2). We found that only models involving Levant_N, Anatolia_N, and either Iran_ChL or Levant_BA_-North passed at $p > 0.05$ (Table 1). To distinguish between Iran_ChL and Levant_BA_North, we added Iran_N to the outgroup set (for a total of $19 = 11 + 8$ outgroups) (Supplementary Figure 4b). Only the model involving Iran_ChL remained plausible. Based on this uniquely fitting qpAdm model we infer the ancestry of Levant_ChL to be the result of a three-way admixture of populations related to Levant_N (57%), Iran_ChL (17%), and Anatolia_N (26%).

**The ancestry of late Levantine Bronze Age populations**. It was striking to us that previously published Bronze Age Levantine samples from the sites of 'Ain Ghazal in present-day Jordan (Levant_BA_South) and Sidon in present-day Lebanon (Levant_BA_North) can be modeled as two-way admixtures, without the Anatolia_N contribution that is required to model the Levant_ChL population[24,26]. This suggests that the Levant_ChL population may not be directly ancestral to these later Bronze Age Levantine populations, because if it were, we would also expect to detect an Anatolia_N component of ancestry. In what follows, we treat Levant_BA_South and Levant_BA_North as separate populations for analysis, since the symmetry statistic $f_4$(Levant_BA_North, Levant_BA_South; A, Chimp) is significant for a number test populations A ($|Z| \geq 3$) (Supplementary Data 5), consistent with the different estimated proportions of Levant_N and Iran_ChL ancestry reported in[24,26].

To test the hypothesis that Levant_ChL may be directly ancestral to the Bronze Age Levantine populations, we attempted to model both Levant_BA_South and Levant_BA_North as two-way admixtures between Levant_ChL and every other ancient population in our dataset, using the base 09NW set of populations as the "Right" outgroups. We also compared these models to the previously published models that used the Levant_N and Iran_ChL populations as sources (Table 2; Supplementary Figure 5; Supplementary Data 6). In the case of Levant_BA_South from 'Ain Ghazal, Jordan, multiple models

were plausible, and thus we returned to the strategy of adding additional "Right" population outgroups that are differentially related to one or more of the "Left" populations (specifically, we added various combinations of Armenia_EBA, Steppe_EMBA, Switzerland_HG, Iran_LN, and Iran_N). Only the model including Levant_N and Iran_ChL remains plausible under all conditions. Thus, we can conclude that groups related to Levant_ChL contributed little ancestry to Levant_BA_South.

We observe a qualitatively different pattern in the Levant_-BA_North samples from Sidon, Lebanon, where models including Levant_ChL paired with either Iran_N, Iran_LN, or Iran_HotuIIIb populations appear to be a significantly better fit than those including Levant_N + Iran_ChL. We largely confirm this result using the "Right" population outgroups defined in Haber et al.[26] (abb. Haber: Ust_Ishim, Kostenki14, MA1, Han, Papuan, Ami, Chuckhi, Karitiana, Mbuti, Switzerland_HG, EHG, WHG, and CHG), although we find that the specific model involving Iran_HotuIIIb no longer works with this "Right" set of populations. Investigating this further, we find that the addition of Anatolia_N in the "Right" outgroup set excludes the model of Levant_N + Iran_ChL favored by Haber et al.[26] These results imply that a population that harbored ancestry more closely related to Levant_ChL than to Levant_N contributed to the Levant_BA_North population, even if it did not contribute detectably to the Levant_BA_South population.

We obtained additional insight by running qpAdm with Levant_BA_South as a target of two-way admixture between Levant_N and Iran_ChL, but now adding Levant_ChL and Anatolia_N to the basic 09NW "Right" set of 11 outgroups. The addition of the Levant_ChL causes the model to fail, indicating that Levant_BA_South and Levant_ChL share ancestry following the separation of both of them from the ancestors of Levant_N and Iran_ChL. Thus, in the past there existed an unsampled population that contributed both to Levant_ChL and to Levant_BA_South, even though Levant_ChL cannot be the direct ancestor of Levant_BA_South because, as described above, it harbors Anatolia_N-related ancestry not present in Levant_BA_South.

**Genetic heterogeneity in the Levantine Bronze Age**. We were concerned that our finding that the Levant_ChL population was a mixture of at least three groups might be an artifact of not having access to samples closely related to the true ancestral populations.

**Table 2 Modeling Levant_BA_South and Levant_BA_North as a mixture of Levant_ChL and an ancient population, A**

| Target | Source left populations | | Outgroup right pops | p Value rank = 2 | Admixture proportions | | Standard error |
|---|---|---|---|---|---|---|---|
| | A | B | | | A | B | |
| *Levant_BA_South* | *Levant_N* | *Iran_ChL* | *09NW* | *9.88E−01* | *0.549* | *0.451* | *0.031* |
| *Levant_BA_South* | *Levant_N* | *Iran_ChL* | *09NWFPY* | *5.14E−01* | *0.571* | *0.429* | *0.026* |
| **Levant_BA_South** | **Levant_N** | **Iran_ChL** | **09NWFPSD** | **1.95E−01** | **0.582** | **0.418** | **0.025** |
| Levant_BA_South | Levant_N | Iran_ChL | 09NWA | 9.94E−01 | 0.55 | 0.45 | 0.027 |
| Levant_BA_South | Levant_N | Iran_ChL | 09NWAZ | 1.39E−02 | 0.601 | 0.399 | 0.026 |
| *Levant_BA_South* | *Levant_ChL* | *CHG* | *09NW* | *5.97E−02* | *0.788* | *0.212* | *0.032* |
| Levant_BA_South | Levant_ChL | CHG | 09NWFPY | 1.82E−03 | 0.812 | 0.188 | 0.024 |
| *Levant_BA_South* | *Levant_ChL* | *Iran_ChL* | *09NW* | *2.00E−01* | *0.714* | *0.286* | *0.04* |
| Levant_BA_South | Levant_ChL | Iran_ChL | 09NWFPY | 3.06E−02 | 0.723 | 0.277 | 0.033 |
| Levant_BA_South | Levant_ChL | Iran_LN | 09NW | 3.53E−01 | 0.717 | 0.283 | 0.039 |
| Levant_BA_South | Levant_ChL | Iran_LN | 09NWFPY | 1.22E−02 | 0.779 | 0.221 | 0.026 |
| *Levant_BA_South* | *Levant_ChL* | *Iran_HotuIIIb* | *09NW* | *2.43E−01* | *0.556* | *0.444* | *0.051* |
| *Levant_BA_South* | *Levant_ChL* | *Iran_HotuIIIb* | *09NWFPSD* | *3.79E−02* | *0.585* | *0.415* | *0.047* |
| *Levant_BA_South* | *Levant_ChL* | *Iran_N* | *09NW* | *4.41E−01* | *0.797* | *0.203* | *0.028* |
| Levant_BA_South | Levant_ChL | Iran_N | 09NWFPSD | 8.00E−04 | 0.853 | 0.147 | 0.075 |
| Levant_BA_North | Levant_N | Iran_ChL | 09NW | 0.003804 | 0.348 | 0.652 | 0.028 |
| *Levant_BA_North* | *Levant_N* | *Iran_ChL* | *Haber* | *0.222705* | *0.518* | *0.482* | *0.04* |
| Levant_BA_North | Levant_N | Iran_ChL | Haber + A | 0.002457 | 0.394 | 0.606 | 0.025 |
| *Levant_BA_North* | *Levant_ChL* | *Iran_LN* | *09NW* | *0.267145* | *0.532* | *0.468* | *0.031* |
| *Levant_BA_North* | *Levant_ChL* | *Iran_LN* | *Haber* | *0.398822* | *0.555* | *0.445* | *0.04* |
| **Levant_BA_North** | **Levant_ChL** | **Iran_LN** | **Haber + A** | **0.455948** | **0.535** | **0.465** | **0.019** |
| *Levant_BA_North* | *Levant_ChL* | *Iran_N* | *09NW* | *0.401157* | *0.63* | *0.37* | *0.024* |
| *Levant_BA_North* | *Levant_ChL* | *Iran_N* | *Haber* | *0.638884* | *0.655* | *0.345* | *0.035* |
| **Levant_BA_North** | **Levant_ChL** | **Iran_N** | **Haber + A** | **0.693801** | **0.638** | **0.362** | **0.015** |
| *Levant_BA_North* | *Levant_ChL* | *Iran_HotuIIIb* | *09NW* | *0.216066* | *0.377* | *0.623* | *0.033* |
| Levant_BA_North | Levant_ChL | Iran_HotuIIIb | Haber | 0.03318 | 0.299 | 0.701 | 0.047 |
| Levant_BA_North | Levant_ChL | Iran_HotuIIIb | Haber + A | 0.007102 | 0.399 | 0.601 | 0.019 |

Note: Populations that produce p values greater than 0.05 with plausible admixture proportions (between 0 and 1) are highlighted in italic. Models that are robust to the maximum number of outgroups are shown in bold

**Table 3 Determining the number of streams of ancestry in the Levant**

| Left pops | Right pops | Rank | Degrees of freedom | Chi squared | p Value |
|---|---|---|---|---|---|
| Levant_N | | 0 | 20 | 190.024 | 1.047e−29 |
| Levant_ChL | 09NW | 1 | 9 | 32.641 | 1.541e−4 |
| Levant_BA_South | | **2** | **0** | **0.000** | **1.000** |
| Levant_N | | 0 | 20 | 399.438 | 2.673e−72 |
| Levant_BA_South | 09NW | **1** | **9** | **6.574** | **0.681** |
| Levant_BA_North | | **2** | **0** | **0.000** | **1.000** |
| Levant_N | | 0 | 20 | 706.552 | 3.221e−135 |
| Levant_BA_South | 09NWZ | 1 | 9 | 28.050 | 1.772e−3 |
| Levant_BA_North | | **2** | **0** | **0.000** | **1.000** |

Note: Models that have a rank that is plausible (i.e., p value of greater than 0.05) are shown in bold. Rank is equal to the minimum number of source populations required to model the "Left" population group relative to the "Right" population group, minus 1 (thus, Rank 2, which is the only working solution for all sets of three "Left" populations, reflects three admixing populations)

One specific possibility we considered is that a single ancestral population admixed into the Levant to contribute to both the Levant_ChL and the Levant_BA_South populations, and that this was an unsampled population on an admixture cline between Anatolia_N and Iran_ChL, explaining why qpAdm requires three source populations to model it. To formally test this hypothesis, we used qpWave[36–38], which determines the minimum number of source populations required to model the relationship between "Left" populations relative to "Right" outgroup populations. Unlike qpAdm, qpWave does not require that populations closely related to the true source populations are available for analysis. Instead it treats all "Left" populations equally, and attempts to determine the minimum number of theoretical source

populations required to model the "Left" population set, relative to the "Right" population outgroups. Therefore, we model the relationship between Levant_N, Levant_ChL, and Levant_BA_-South as "Left" populations, relative to the 09NW "Right" outgroup populations (Table 3). We find that a minimum of three source populations continues to be required to model the ancestry of these Levantine populations, supporting a model in which at least three separate sources of ancestry are present in the Levant between the Neolithic, Chalcolithic, and Bronze Age.

We applied qpWave again, replacing Levant_ChL with Levant_BA_North, and found that the minimum number of source populations is only two. However, when we include the Levant_ChL population as an additional outgroup, three source

populations are again required. This suggests that in the absence of the data from Levant_ChL there is insufficient statistical leverage to detect Anatolian-related ancestry that is truly present in admixed form in the Levant_BA_North population (data from the Levant_ChL population makes it possible to detect this ancestry). This may explain why Haber et al.[26] did not detect the Anatolian Neolithic-related admixture in Levant_BA_North.

**Biologically important mutations in the Peqi'in population.** This study nearly doubles the number of individuals with genome-wide data from the ancient Levant. Measured in terms of the average coverage at SNPs, the increase is even more pronounced due to the higher quality of the data reported here than in previous studies of ancient Near Easterners[24,26]. Thus, the present study substantially increases the power to analyze the change in frequencies of alleles known to be biologically important.

We leveraged our data to examine the change in frequency of SNP alleles known to be related to metabolism, pigmentation, disease susceptibility, immunity, and inflammation in the Levant_ChL population, considered in relation to allele frequencies in the Levant_N, Levant_BA_North, Levant_BA_South, Anatolia_N and Iran_ChL populations and present-day pools of African (AFR), East Asian (EAS), and European (EUR) ancestry in the 1000 Genomes Project Phase 3 dataset[39] (Supplementary Data 7).

We highlight three findings of interest. First, an allele (G) at *rs12913832* near the *OCA2* gene, with a proven association to blue eye color in individuals of European descent[40], has an estimated alternative allele frequency of 49% in the Levant_ChL population, suggesting that the blue-eyed phenotype was common in the Levant_ChL.

Second, an allele at *rs1426654* in the *SLC24A5* gene which is one of the most important determinants of light pigmentation in West Eurasians[41] is fixed for the derived allele (A) in the Levant_ChL population suggesting that a light skinned phenotype may have been common in this population, although any inferences about skin pigmentation based on allele frequencies observed at a single site need to be viewed with caution[42].

Third, an allele (G) at *rs6903823* in the *ZKSCAN3* and *ZSCAN31* genes which is absent in all early agriculturalists reported to date (Levant_N, Anatolia_N, Iran_N) and that has been argued to have been under positive selection by Mathieson et al.[31], occurs with an estimated frequency of 20% in the Levant_ChL, 17% in the Levant_BA_South, and 15% in the Iran_ChL populations, while it is absent in all other populations. This suggests that the allele was rising in frequency in Chalcolithic and Bronze Age Near Eastern populations at the same time as it was rising in frequency in Europe.

## Discussion

The Chalcolithic period in the Levant witnessed major cultural transformations in virtually all areas of culture, including craft production, mortuary and ritual practices, settlement patterns, and iconographic and symbolic expression[43]. The current study provides insight into a long-standing debate in the prehistory of the Levant, implying that the emergence of the Chalcolithic material culture was associated with population movement and turnover.

The quality of ancient DNA obtained from the Peqi'in Cave samples is excellent relative to other sites in the Near East. We hypothesize that the exceptional preservation is due to two factors. First, the targeted sampling of ancient DNA from the petrous portion of the temporal bone makes it possible to obtain high-quality ancient DNA from previously inaccessible

geographic regions[24,27,44,45]. Secondly, the environment of Peqi'in Cave is likely to be favorable for DNA preservation. The skeletal remains—either stored in ossuaries or laid in the ground—were quickly covered by a limestone crust, isolating them from their immediate surroundings and protecting them from acidic conditions that are known to be damaging to DNA.

We find that the individuals buried in Peqi'in Cave represent a relatively genetically homogenous population. This homogeneity is evident not only in the genome-wide analyses but also in the fact that most of the male individuals (nine out of ten) belong to the Y-chromosome haplogroup T (see Supplementary Table 1), a lineage thought to have diversified in the Near East[46]. This finding contrasts with both earlier (Neolithic and Epipaleolithic) Levantine populations, which were dominated by haplogroup E[24], and later Bronze Age individuals, all of whom belonged to haplogroup J[24,26].

Our finding that the Levant_ChL population can be well-modeled as a three-way admixture between Levant_N (57%), Anatolia_N (26%), and Iran_ChL (17%), while the Levant_BA_South can be modeled as a mixture of Levant_N (58%) and Iran_ChL (42%), but has little if any additional Anatolia_N-related ancestry, can only be explained by multiple episodes of population movement. The presence of Iran_ChL-related ancestry in both populations – but not in the earlier Levant_N – suggests a history of spread into the Levant of peoples related to Iranian agriculturalists, which must have occurred at least by the time of the Chalcolithic. The Anatolian_N component present in the Levant_ChL but not in the Levant_BA_South sample suggests that there was also a separate spread of Anatolian-related people into the region. The Levant_BA_South population may thus represent a remnant of a population that formed after an initial spread of Iran_ChL-related ancestry into the Levant that was not affected by the spread of an Anatolia_N-related population, or perhaps a reintroduction of a population without Anatolia_N-related ancestry to the region. We additionally find that the Levant_ChL population does not serve as a likely source of the Levantine-related ancestry in present-day East African populations (see Supplementary Note 4)[24].

These genetic results have striking correlates to material culture changes in the archaeological record. The archaeological finds at Peqi'in Cave share distinctive characteristics with other Chalcolithic sites, both to the north and south, including secondary burial in ossuaries with iconographic and geometric designs. It has been suggested that some Late Chalcolithic burial customs, artifacts and motifs may have had their origin in earlier Neolithic traditions in Anatolia and northern Mesopotamia[8,13,47]. Some of the artistic expressions have been related to finds and ideas and to later religious concepts such as the gods Inanna and Dumuzi from these more northern regions[6,8,47–50]. The knowledge and resources required to produce metallurgical artifacts in the Levant have also been hypothesized to come from the north[11,51].

Our finding of genetic discontinuity between the Chalcolithic and Early Bronze Age periods also resonates with aspects of the archeological record marked by dramatic changes in settlement patterns[43], large-scale abandonment of sites[52–55], many fewer items with symbolic meaning, and shifts in burial practices, including the disappearance of secondary burial in ossuaries[56–59]. This supports the view that profound cultural upheaval, leading to the extinction of populations, was associated with the collapse of the Chalcolithic culture in this region[18,60–64].

These ancient DNA results reveal a relatively genetically homogeneous population in Peqi'in. We show that the movements of people within the region of the southern Levant were remarkably dynamic, with some populations, such as the one buried at Peqi'in, being formed in part by exogenous influences.

This study also provides a case-study relevant beyond the Levant, showing how combined analysis of genetic and archaeological data can provide rich information about the mechanism of change in past societies.

## Methods

**Data generation**. Peqi'in burial Cave was excavated under the auspices of the Israel Antiquities Authority (Permit no. 2297/1995). We screened 46 human skeletal elements from Peqi'in Cave, of which 37 were petrous fragments of the temporal bone. We prepared between 15 and 114 mg of bone powder for each sample by drilling from a compact part of the sample after surface cleaning using a Dremel tool or by drilling into the inner ear part of the petrous portion of the temporal bone[27]. We extracted DNA using a silica-column-based extraction protocol that has been optimized for ancient DNA extraction[28], modifying the protocols by replacing the MinElute column assembly with a preassembled spin column device as in Korlević et al.[65]. We added 1.5 mL of extraction buffer (0.45 M EDTA, pH 8.0 (BioExpress), 0.05% Proteinase K (Sigma)) to the bone powder and incubated in 2.0 mL tubes at 37 °C overnight, while rotating. Following incubation, we centrifuged the samples at maximum speed for 2 min, and added 13 mL of binding buffer (5 M GuHCl (Sigma), 40% Isopropanol (Sigma), 400 μg Sodium Acetate (Sigma), pH 5.2 (Sigma)) to the supernatant. We transferred the mixture to a High Pure Extender from a Viral Nucleic Acid Large Volume Kit (Roche) and centrifuged at 2000×g until all liquid disappeared from the funnel. We detached the silica column from the funnel, placed it in a fresh 2 mL collection tube, and spun for 1 min at 8000×g. We performed two washes by adding 700 μL PE buffer (Qiagen) to the columns, and spun at 8000×g for 30 s, replacing the collection tube after each wash. We performed a dry spin at maximum speed for 1 min, and then replaced the collection tube. We removed the DNA eluate from the column by adding 45 μL of TTE (10 mM Tris-HCl, pH 8.0 (ThermoFisher), 1 mM EDTA, pH 9.0 (BioExpress), 0.05% Tween-20 (Sigma)) to the silica matrix, incubating for 5 min, and then spun at maximum speed for 1 min. We repeated this step until we obtained a total volume of 90 μL. For reattempts of one of the samples, we washed the powder with 1 mL 0.5% bleach (incubating for 15 min), followed by three washes with 1 mL water (incubating 3 min), prior to DNA extraction as described in Korlević et al.[65] (see Supplementary Data 1), and prepared libraries using partial UDG treatment[29] (the library protocols varied slightly over the course of data generation, see Supplementary Data 1). We added 30 μL of extract to the USER treatment mixture (1× Buffer Tango (ThermoFisher), 100 μM dNTP Mix (ThermoFisher), 1 mM ATP (ThermoFisher), 0.06 U/μL USER enzyme (NEB)), and incubated the reaction at 37 °C for 30 min. We inhibited the UDG enzyme by adding Uracil Glycosylase Inhibitor (0.12 U/μL; NEB) to the mix and incubating for a further 30 min at 37 °C. We then performed blunt end repair on the samples by adding T4 PNK (0.5 U/μL; ThermoFisher) and T4 Polymerase (90.1 U/μL; ThermoFisher) to the mixture and incubating for 15 min at 25 °C, followed by 5 min at 12 °C. We cleaned the reactions up using a MinElute PCR purification kit, adding five volumes of PB buffer to the reaction mixture, transferring to a collection tube, and spinning for 30 s at 3300×g. We discarded the liquid and washed twice by adding 700 μL of PE buffer to the column, centrifuging for 30 s at max speed, and discarding the collection tube, followed by a dry spin for 1 min at maximum speed. We eluted the samples in 18 μL of 10 mM Tris-HCl (ThermoFisher), which we added to the silica membrane and allowed to sit for 5 min, followed by centrifuging for 1 min at maximum speed. We ligated unique adapters to the molecules in each sample by incubating the sample mixture in a ligation reaction mixture (1× T4 DNA ligase buffer (ThermoFisher), 5% PEG-4000 (ThermoFisher), 0.25 μM P5-adapter (see ref. [29] for suggested preparation information), 0.25 μM P7 adapter (see ref. [29] for suggested preparation information), 0.125 U/μL T4 DNA ligase (ThermoFisher)) for 30 min at room temperature. We cleaned up the ligation mixture using the clean-up procedure described above, eluting in 20 μL 10 mM Tris-HCl. We filled in the ligated adapters by adding a fill-in reaction mixture (1× ThermoPol buffer (NEB), 250 μM dNTP Mix (ThermoFisher), 0.4 U/μL Bst Polymerase, large fragment (NEB)) to the ligation product, and incubating at 37 °C for 20 min, followed by 80 °C for 20 min. Finally, we amplified the libraries via PCR by adding 39 μL of the fill-in reaction product to the PCR reaction mixture (1× Pfu Turbo Cx Reaction Buffer (Agilent Technologies), 0.4 μM PreHyb-F (5′-CTTTCCCTA-CACGACGCTCTTC-3′), 0.4 μM PreHyb-R (5′-GTGACTGGAGTTCAGA CGTGTGCT-3′), 0.2 mM dNTP Mix (ThermoFisher), 5U Pfu Turbo Cx Hotstart DNA Polymerase (Agilent Technologies)). We divided each reaction into 50 μL aliquots and placed in a sealed PCR plate. We performed initial denaturation of the samples for 2 min at 95 °C, followed by 30 cycles at 95 °C for 30 s, 55 °C for 30 s, 72 °C for 1 min, and performed a final extension at 72 °C for 10 min.

We screened all libraries for authentic DNA by enriching for the mitochondrial genome and 50 nuclear target loci, followed by sequencing on an Illumina NextSeq500 instrument for 2 × 75 cycles and 2 × 7 cycles to read out the indices. We enriched promising libraries for approximately 1.2 M SNPs as described in refs. [31,36,66,] and then sequenced on a NextSeq500 sequencer using 75 base pair paired-end sequences. During computational processing, we initially stripped identifying oligonucleotide sequences and adapters, separating individual samples

from pooled captures by their identifying 7 base pair indices at the 5′ and 3′ ends of reads, and requiring matches to sample-specific barcodes appended directly to the sequence fragments, allowing no more than one mismatch per index/barcode. We used SeqPrep[67] to strip adapters and also to merge paired end reads into single sequences by requiring a minimum of 15 base pair overlap (allowing up to one mismatch), using the highest quality base in the merged region where there was a conflict. We used samse in bwa (v0.6.1)[68] to align reads. For the mitochondrial DNA enrichment experiment we aligned to the RSRS mitochondrial genome[69]. For the whole-genome enrichment experiment we aligned to the hg19 reference genome. We identified duplicate sequences as ones with the same start and end positions and orientation and also identical barcode pairs, and retained the highest quality sequence from each duplicate. We made pseudo-haploid SNP calls for each position using a randomly chosen sequence covering each targeted site, stripping the two bases at the ends of each sequence to remove deaminated mutations, and requiring a minimum mapping quality (MAPQ ≥ 10), and restricting to sites with a minimum base quality (≥20).

We assessed the quality of each library at the screening stage using three standard methods for determining ancient DNA authenticity. First, we analyzed mitochondrial genome data to determine the rate of matching to the consensus sequence, using contamMix[30]. Second, we restricted to samples in which the rate of C-to-T substitutions in terminal nucleotides was at least 3%, as expected for genuine ancient DNA using the partial UDG treatment protocol[29]. Finally, we used the ANGSD software to obtain a conservative estimate of contamination in the X-chromosome of individuals determined to be male based on the rate of polymorphism on X-chromosome sequences (males have only a single X-chromosome and so are not expected to show polymorphism); we excluded libraries with X-contamination estimates greater than 1.5%[32]. For samples where multiple libraries were produced for a single individual, we merged libraries that passed quality control, and obtained new pseudo-haploid SNP calls.

We determined mitochondrial DNA haplogroups using the tool haplogrep2[70], using a consensus sequence built from reads enriched for the mitochondrial genome, restricting to damaged reads using PMDtools[71] (pmdscore ≥ 3), and trimming 5 bases from each end to greatly reduce the error rate due to deamination.

Ancient DNA presents challenges in the assignment of Y-chromosome haplogroups due to the chance that there may be contamination, DNA damage or missing data present in them. In order to assign Y haplogroups to our data, we used a modified version of the procedure used in the analysis of modern Y chromosomes in the 1000 Genomes Project[72], which uses a breadth-first search to traverse the Y-chromosome tree. We made our calls on the ISOGG tree from 04.01.2016 [http://isogg.org], and modified the caller to output derived and ancestral allele calls for each informative position on the tree. We then assigned a score to each of the reference haplogroups by counting the number of mismatches in the number of observed derived alleles on that branch and down-weighted derived mutations that were transitions to 1/3 that of transversions to account for DNA damage related errors. We assigned the sample to the reference haplogroup with the closest match based on this score. While we endeavored to produce a call on each sample, we note that samples with fewer than 100,000 SNPs have too little data to confidently identify the correct haplogroup, and we encourage caution when interpreting these results.

The data from the 22 samples that passed contamination and quality control tests are reported in Supplementary Table 1, with an average of 0.97× coverage on the 1240 k SNP targets, and an average of 358,313 SNPs covered at least once. A by-library table describing the screening results is reported in Supplementary Data 1. We excluded two individuals from further analysis, as the genetic patterns observed using the method described in Kuhn et al.[73]. Showed that they were first-degree relatives of higher coverage samples in the dataset. We restricted data from sample I1183 to include only sequences with evidence of C-to-T substitution in order to minimize contamination which was evident in the full data from these samples.

We combined the newly reported data with existing data from Lazaridis et al.[24] and Haber et al.[26], using the mergeit program of EIGENSOFT[33]. The resulting datasets, referred to as HO + and HOIll+, contain the 20 new unrelated samples combined with HO and HOIll from Lazaridis et al.[24] and 5 ancient samples from Sidon, Lebanon (population name: Levant_BA_North) from Haber et al.[26], respectively. HO+ includes data from 2891 modern and ancient individuals at 591,642 SNPs, and *HOIll+* includes data from 306 ancient individuals at 1,054,637 SNPs.

**Principal component analysis**. We performed PCA on the HO+ dataset using smartpca[33]. We used a total of 984 present-day individuals for PCA, and projected the 306 ancient samples. We used default parameters with lsqproject: YES and numoutlieriter: 0 settings. We estimated $F_{ST}$ using smartpca for the 21 ancient Near Eastern populations made up of more than one individual and 8 modern populations using default parameters, with inbreed: YES and fstonly: YES (Supplementary Figure 1). We ran analyses using the HO+ dataset.

**Admixture**. We carried out ADMIXTURE analysis[34] on the HO+ dataset. Prior to analyses, we pruned SNPs in strong linkage disequilibrium with each other using PLINK[74] using the parameters—indep-pairwise 200 25 0.4. We performed ADMIXTURE analysis on the 3,00,885 SNPs remaining in the pruned dataset. For

each value of $k$ between 2 and 14, we performed 20 replicate analyses, and we retained the highest likelihood replicate for each $k$.

**Conditional heterozygosity**. We computed conditional heterozygosity for each ancient Levantine population using popstats[75]. For this analysis we used the HO+ dataset, restricting to SNP sites ascertained from a single Yoruba individual and to transversion SNPs, as described in Skoglund et al.[44].

**$f$-statistics**. We computed $f_4$-statistics using the qpDstat program in ADMIX-TOOLS[35], with default parameters, and $f_4$ mode:YES. We computed $f_3$-statistics using the qp3Pop program in ADMIXTOOLS[35], using default parameters, with inbreed: YES. We ran all analyses using the HOIll+ dataset, except for the statistic $f_4$(Levant_BA_North, Levant_BA_South; A, Chimp), which we ran on the HO+ dataset.

**qpAdm**. We estimated proportions of ancestry in the Levant_ChL population using the qpAdm methodology, with parameters allsnps: YES and details:YES[36]. We tested both 2- and 3-way admixtures between ancient "Left" populations from the HOIll+ dataset. We used the 09NW populations defined in Lazaridis et al.[24] as preliminary outgroups. We selected additional outgroups based on the statistics $f_4$(Anatolia_N, Europe_EN; A, Chimpanzee) and $f_4$(Levant_BA_North, Iran_ChL; A, Chimpanzee), and we repeated qpAdm with each additional outgroup added into the "Right" list until all but one admixture model was eliminated.

We used qpAdm to determine whether the Levant_BA_South and Levant_BA_North populations could be modeled using Levant_ChL as a source population. We tested 2-way admixtures between Levant_ChL and every other ancient "Left" population from the HOIll+ dataset. We also tested the "Left" populations Levant_N and Iran_ChL. We used the 09NW "Right" populations as preliminary outgroup populations, and confirmed our findings for Levant_BA_North using the outgroups defined in Haber et al.[26]. We added additional outgroups to further differentiate between plausible models, and repeated qpAdm analysis until all but one candidate admixture model was eliminated.

**qpWave**. We computed the minimum number of streams of ancestry required to model two sets of three Levantine populations (set [1] Levant_N, Levant_ChL, and Levant_BA_South, set [2] Levant_N, Levant_BA_South, Levant_BA_North) using the qpWave[37,38] methodology with parameter allsnps:YES.

**Allele frequency comparisons**. We examined the frequencies of SNPs associated with phenotypically important functions in the categories of metabolism, pigmentation, disease susceptibility, immunity, and inflammation in Levant_ChL in conjunction with the Levant_N, Levant_BA_North, Levant_BA_South, Anatolia_N and Iran_ChL populations, with allele frequencies for three pooled continental populations (AFR, EAS, EUR) in Phase 3 the 1000 Genomes Project reported where available. We computed allele frequencies at each site of interest by computing the likelihood of the population reference allele frequency given the data, using a method established in Mathieson et al.[31]. For each population of size, $N$, we observe $R_i$ sequences that possess the reference allele out of a total $T_i$ sequences. The likelihood of the reference allele frequency, $p$, in each population given the data $D = \{X, N, R_i, T_i\}$ is $L(p;D) = \prod_{i=1}^{N} \{p^2 B(R_i, T_i, 1 - \varepsilon) + 2p(1 - p)B(R_i, T_i, 0.5) + (1 - p)^2 B(R_i, T_i, \varepsilon)\}$ where $B(k, n, p) = \binom{n}{k} p^k (1 - p)^{n-k}$ is the binomial probability distribution, and $\varepsilon$ is a small probability of error, which we set to 0.001 for our calculations. We estimated allele frequencies by maximizing the likelihood numerically for each population.

**Data availability**. The aligned sequences are available through the European Nucleotide Archive under accession number PRJEB27215. Genotype datasets used in analysis are available at https://reich.hms.harvard.edu/datasets.

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

## Acknowledgments

Peqi'in burial Cave was excavated under the auspices of the Israel Antiquities Authority. E.H. was supported by a graduate student fellowship from the Max Planck–Harvard Research Center for the Archaeoscience of the Ancient Mediterranean (MHAAM). D.R. was supported by the U.S. National Science Foundation HOMINID grant BCS-1032255, the U.S. National Institutes of Health grant GM100233, by an Allen Discovery Center grant, and is an investigator of the Howard Hughes Medical Institute. The anthropological study was supported by the Dan David Foundation. We thank Vagheesh Narasimhan for generating and describing Y-chromosome haplogroup calls. We thank Ariel Pokhojaev for creating the map image used in Fig. 1a. We thank John Wakeley for critical comments.

## Author contributions

H.M, I.H., and D.R. conceived the study. D.R. supervised the ancient DNA work, sequencing, and data analysis. H.M, D.S, R.S., and I.H. assembled, studied, or described the archaeological material. E.H., H.M., N.R., K.S, and S.N performed or supervised wet laboratory work. S.M performed bioinformatics analyses. E.H. performed population genetics analyses, with I.L. and N.P. providing guidance. E.H., H.M, I.H, and D.R wrote the manuscript with input from all co-authors.

## Additional information

**Competing interests:** The authors declare no competing interests.

