## [Peer Review File · Nature Communications]

Reviewers' comments:

Reviewer #1 (Remarks to the Author):

Harney et al review

In this manuscript Harney et al. present new genome-wide SNP data from 22 individuals from the Levant (Israel) dating to the Chalcolithic. They compare these samples primarily to other Levantine, Anatolian and Iranian genomes from the Neolithic through to the Bronze age, with the main aim of exploring whether at least two archaeological shifts that occur between these two points correlate with shifts in genetic ancestry, which may implicate population turnover and migration from outside the region. The authors conclude that Chalcolithic Levant populations as represented by this one specific site show genetic evidence of being a mixture of early Neolithic farmers from the region, Iranian Chalcolithic, and somewhat surprisingly, Anatolian Neolithic farmers, thus supporting the conclusion of migration. The authors also suggest only continuity with some Bronze Age populations from the same area.

This data produced in this paper is unique with regard to the scale of samples from a single site, especially one so old. The authors use a series of established but powerful methods that they have developed themselves to perform the demographic analysis. I imagine both the results and data will be of interest to those working on ancient DNA, population geneticists, archaeologists and anthropologists, given how important this part of the world is to understanding human prehistory out of Africa. In general the results seem robust but I do have some major issues that need addressing.

I note that I am not qualified to evaluate the archaeological claims.

Major Issues:

1. The primary evidence of migration into the area and population replacement suggested by the authors between the Neolithic and Chalcolithic (and then the Bronze age) that would explain changes in archaeological complexity hinges on new ancestry components entering the Chalcolithic Levant, and thus modeling them as a mixture of Neolithic Levant and other sources. However, I worry to what extent treating the Neolithic Levant samples as a single population is influencing some of these results. Is this really justified? My understanding is that the Neolithic Levant samples come from two locations and from multiple time periods during the Neolithic. There is a clear overlap for some samples between the Neolithic and Chalcolithic Levant in the PCA. In addition, in the Admixture analysis there are some Neolithic Levant individuals that appear to match Chalcolithic individuals, with the same ratio of blue and green component, while some Neolithic samples have zero green component that is so prominent in Neolithic Iranian populations. There are even two individuals that have what looks like a component (red) found only in European and Anatolian Neolithic samples. Could it be that Chalcolithic Levant (and perhaps only individuals from this particular site, rather than the region in general) simply emerged from one part of a

structured Neolithic Levant population, and that no major migration into the region is needed to explain the results (particularly from the Iranian side)? I'd would like to see the authors test to what extent population structure is influencing the results, either redoing the various qpAdmix analysis with individual Neolithic Levant individuals or grouping by sampling location and time period. Perhaps the original results will still emerge, but it should be tested.

2. The qpAdmix (and qWave) sections that form the bulk of the results after the PCA and Admixture analysis are very difficult to read. It is hard work to follow the various tests being conducted and how they fit in to the larger picture. I know the authors developed these methods and thus can construct elaborate designs using various combinations of samples, but for people outside the field in particular, this section will be very hard to work through. Some figures showing the various models being tested would be useful, even if they get put in the supplementary material.

3. One of the most unique aspects of this study is the generation of genomic data from a large sample from a single site, something the authors themselves highlight. However, very little is actually done with this, other than to say they are "homogenous"? This is disappointing. There does not seem to be much they have done with 21 samples that they could not have done with 1 or two samples. While I know this is only SNP-capture data, are there any inferences that could be made about population diversity or levels of inbreeding. It seems a waste. Also, what about population frequencies of phenotypic markers that I believe are within the captured SNP set based on a previous manuscript by this group (Mathieson et al. 2015)?

Specific Minor Issues.

Line 47: "One hypothesis is that the Chalcolithic culture in the region was spread by immigrants from the north, based on similarities in decorative methods"

Please be more specific about "north". There is a lot of land up there!

Line 58: "It has been estimated that the burial cave contained up to 600 individuals"

Given the dimensions of the cave given above, this seems a lot. Is there a picture or figure of the site we could see?

Line 60: "Direct radiocarbon dating suggests that the cave was in use throughout the Late Chalcolithic period (c. 4500–3950 BCE), functioning as a secondary burial cemetery...suggests that it functioned as a central cemetery for the settlements located in the western Upper Galilee"

This is confusing, is it a secondary burial cemetery or a central cemetery? Please clarify (perhaps it can be both, though I'm not immediately sure how")

Line 103: "The dataset is of extraordinarily quality"

Though I very much appreciate the value of the data, this is a bit hyperbolic. It is only "extraordinary" if placed into some context with samples from similar time periods. Please either rephrase or contextualize.

Line 118: "The Levant_ChL cluster falls in between the clusters corresponding to the Levant Neolithic (Levant_N) and Levant Bronze Age (including samples from 'Ain Ghazal, Jordan (abb.: Levant_BA_South) and Sidon, Lebanon (abb.: Levant_BA_North))."

From my examination of the PCA, it looks like the Levant ChL significantly overlap with Levant_N rather than being in between. Also, are all the other ancient (and even modern) samples from Europe necessary for this PCA? It would be much easier to understand the pattern if we could zoom into the region and samples of interest. Everything else is a bit of a distraction.

Line 128: "Figure 1d shows the ADMIXTURE results for the ancient individuals assuming K=11 clusters (we selected this number because it maximizes ancestry components that are correlated to ancient populations from the Levant, from Iran, and European hunter-gatherers)"

Could the relevant individuals be grouped together a little bit better, in particular Iranian Neolithic samples being close to the Levant samples? Also, the point must be made in the main text (see above) that some Levant Neolithic samples appear to show the same pattern as the Levant ChL samples (perhaps corresponding with their overlap in the PCA).

Line 144: "Figure 2a shows that Levant_N shares more alleles with the Levant_ChL than it does with any other population, as the statistic is always positive. However, Levant_ChL is also not directly descended from Levant_N"

As described above, I assume the Levant N samples are grouped together and analyzed simultaneously in this analysis. How does variation with Levant N affect these analyses (what if for example each Levant N sample is analyzed individually)? What is the impact of population structure within Levant N on the model, methods and interpretation?). Also, not all these comparisons appear to be significant, in particular for European early and Anatolian Neolithic.

Line 167: "To search for models that fit the ancestry of the Levant_ChL, we chose as Left populations Levant_ChL along with various subsets of the 5 populations that are geographically and temporally closest to the Levant_ChL (Anatolia_N, Anatolia_ChL, Armenia_ChL, Iran_ChL, Iran_N, Levant_N), in conjunction with other ancient populations." What does the "in conjunction" refer to in the analysis framework described in the previous paragraph? The ancestral populations? A bit confusing.

Line 220: "particularly other ancient Levantine populations (Supplementary Table 6)"
Is Supp Table 6 correct, it just has one row in it?

Line 281: "suggesting that the emergence of the Chalcolithic material culture was associated with population migration and turnover....The presence of Iran_ChL-related ancestry in both populations – but not in the earlier Levant_N – suggests migration into the Levant of populations related to Iranian farmers, which must have occurred at least by the time of the Chalcolithic. The Anatolian_N component present in the Levant_ChL but not in the Levant_BA_South sample suggests that there was also a separate migration of Anatolian-related people into the region."

See major issue about population structure on whether these inferences are supported.

Line 386: "We estimated FST using smartpca for the 21 ancient Near Eastern populations made"

The results of this are never mentioned.

Reviewer #2 (Remarks to the Author):

Not being an expert on DNA research, I will limit my comments to the archaeological aspects of the paper.

The southern Levant Ghassulian culture of the Chalcolithic period (ca. 4,500-3900 BC cal.) features many unique and innovative cultural attributes, as the authors indicate. The copper industry, for one, especially the use of the "lost wax" technique, is probably the earliest advanced metallurgy known. Since its discovery in the late 1920s, the thorny questions of the Ghassulian origin and its termination are central to the research of this time span. Until now the attempts to address this question were based essentially on "subjective" comparisons of iconography and artifacts' shapes. The proposed paper offers a totally different "objective" answers on the basis of comparing ancient DNA extracted from human bones from the cave of Peki'in in the Upper Galilee, Israel, to earlier and later populations from the southern Levant, as well as to populations from Iran and Anatolia.

The proposed paper advocates that the long debate is now settled: the ancient DNA shows that the uniqueness of the Ghassulian is due to an intrusive cultural entity. The genetic evidence indicates complex movements and local populations turnovers. Thus, it is argued that two waves of migrations, from Iran and from Anatolia, triggered the inception of the Ghassulian. Concerning the end of this culture, the authors argue that the succeeding Bronze Age population shows different genetic attributes and therefore represents a discontinuity, well reflected in the material culture that differs from the previous period.

The idea that the Ghassulian Chalcolithic was an intrusive culture has already been suggested in the past on the basis of equivocal typological grounds. However, the intrusion model is now supported by ancient DNA, and this is a novelty. The results of this study are of interest to a number of research communities including archaeologists, ancient DNA geneticists, anthropologists and historians of technology.

I recommend to publish the paper, but I suggest to consider corrections and comments added to the text which I hereby attach. I've also found mistakes in the list of references concerning the archaeological aspect of the paper. I suggest a number of corrections in the attached file and recommend a careful re-checking of the list.

Reviewer #3 (Remarks to the Author):

This is a report of a successful genomic analysis of a population sample from an interesting

archaeological context - the Levantine Chalcolithic. Substantial genomewide data from 22 individuals are obtained, and these allow new inference about the complexity of migration and admixture in the ancient Near East. Some positives about this work are:

- the densities of coverage of the targeted SNPs are unusually good for this region and time. This makes this a valuable contribution for further studies. Is it worth speculating on why? Was this a limestone context, free from water disturbance etc.
- The wider archaeological context is well discussed and referenced.
- The implication of different ancestral components and therefore discontinuity with earlier and later horizons is novel and interesting, pointing to population flux
- The analysis seems sound and draws on existing established methods.

As such I judge that this work has the basis for publication in the journal.

Some comments:

The inference relies strongly upon qpadm - is there merit in addressing this question using alternative approaches, eg Treemix?

This unveils a new population sample from a region of strong interest, is there scope for asking if this population has made a specific contribution to later populations, eg in North Africa.

The unusually strong recovery of genomewide data should make possible the assessment of how variation maps across the genome. Are there any regions showing outlying diversity that may be signals of selection in this population?

A small quibble, the reference given in the main text for the petrous bone's aDNA preservation qualities is Pinhasi et al. whereas in fact the primary reference is Gamba et al.

Summary of Comments from Reviewer 2 on Article

Page: 3

Author: redacted Subject: Sticky Note Date: 12/17/2017 5:45:51 PM -05'00'

All the defining attributes listed below characterize a segment of the "Late Chalcolithic" period. There are parts in the north and the south of the southern Levant, in which none of these distinctive characteristics are found.

The listed attributes characterize the Chalcolithic cultural entity most authorities refer to as "Ghassulian".

I suggest that the terms Ghassulian be mentioned in the text, at least in this introductory section.

Author: redacted Subject: Highlight Date: 12/16/2017 9:06:46 AM -05'00'

Author: redacted Subject: Sticky Note Date: 12/17/2017 5:46:15 PM -05'00'

, and the use of the "lost wax" technique,

Author: redacted Subject: Highlight Date: 12/16/2017 9:06:55 AM -05'00'

Author: redacted Subject: Sticky Note Date: 12/17/2017 5:47:14 PM -05'00'

Perrot in his 1955 paper (reference 12) discusses the origin of the "Beersheba culture", and suggests that it originated in Trans-Jordan (ibid pp. 185-188). Concerning the other Chalcolithic sites, he claims that there are sites "...of local origin..." and a group of sites which "We do not know its immediate origins." Thus, this reference does not support the claim for a northern origin and should be removed.

The same applies to reference 13. A northern origin of the "Ghassoulien" is not mentioned.

Author: redacted Subject: Sticky Note Date: 12/17/2017 5:47:34 PM -05'00'

Kenyon (1979: p. 63) suggest that the Ghassulians "...must have come from the east or north-east...".

Author: redacted Subject: Highlight Date: 12/16/2017 10:07:14 AM -05'00'

Author: redacted Subject: Highlight Date: 12/16/2017 9:07:48 AM -05'00'

Author: redacted Subject: Sticky Note Date: 12/16/2017 10:23:03 AM -05'00'

The "burial jars" of Peki'in are ossuaries.

Author: redacted Subject: Highlight Date: 12/16/2017 10:08:37 AM -05'00'

Page: 4

 Author: redacted Subject: Sticky Note Date: 12/17/2017 5:50:50 PM -05'00'

A problematic statement. The Peki'in cultural assemblages had strong cultural affinities with the Ghassulian culture (off-sites cemeteries in ossuaries, similar iconography, the same pottery vessel types, basalt bowls, violin-shaped figurines, copper artifacts etc.). There are also affinities with the Golan Chalcolithic, pottery of which is present at the cave.

If the Peki'in cultural affinities are enigmatic, how do they contribute to understanding the genetic origins of the what the authors label "Late Chalcolithic" (Ghassulian)?

 Author: redacted Subject: Highlight Date: 12/16/2017 10:12:24 AM -05'00'

Page: 6

Author: redacted Subject: Sticky Note Date: 12/16/2017 10:46:28 AM -05'00'

3900?

Author: redacted Subject: Highlight Date: 12/16/2017 10:46:28 AM -05'00'

Page: 15

Author: redacted Subject: Sticky Note Date: 12/16/2017 10:56:36 AM -05'00'
Contradicts the statements in lines 64-65, concerning the "enigmatic" cultural affinities of Peki'in (see my comment there).

Author: redacted Subject: Highlight Date: 12/16/2017 10:52:50 AM -05'00'

Author: redacted Subject: Highlight Date: 12/16/2017 10:56:14 AM -05'00'

Author: redacted Subject: Sticky Note Date: 12/17/2017 5:51:27 PM -05'00'
Add: de Vaux 1970: 529-530 (reference 15).

Author: redacted Subject: Highlight Date: 12/16/2017 11:39:25 AM -05'00'

Page: 23

Author: redacted Subject: Sticky Note Date: 12/16/2017 12:06:22 PM -05'00'
Italics

Author: redacted Subject: Highlight Date: 12/16/2017 11:49:06 AM -05'00'

Author: redacted Subject: Sticky Note Date: 12/16/2017 11:50:52 AM -05'00'
Diacritics.

Author: redacted Subject: Highlight Date: 12/16/2017 11:50:10 AM -05'00'

Author: redacted Subject: Sticky Note Date: 12/16/2017 11:51:25 AM -05'00'
Italics

Author: redacted Subject: Highlight Date: 12/16/2017 11:51:01 AM -05'00'

Author: redacted Subject: Highlight Date: 12/16/2017 11:51:47 AM -05'00'

Author: redacted Subject: Highlight Date: 12/16/2017 11:51:32 AM -05'00'

Author: redacted Subject: Highlight Date: 12/16/2017 11:52:23 AM -05'00'

Author: redacted Subject: Highlight Date: 12/16/2017 11:52:30 AM -05'00'

Page: 25

Author: redacted Subject: Sticky Note Date: 12/16/2017 11:57:42 AM -05'00'
(Hebrew).

Author: redacted Subject: Sticky Note Date: 12/16/2017 12:00:45 PM -05'00'
Vardi, J. and Gilead, I.

Author: redacted Subject: Highlight Date: 12/16/2017 11:59:48 AM -05'00'

Author: redacted Subject: Sticky Note Date: 12/16/2017 12:01:43 PM -05'00'
Braun, E. & Roux, V.

Author: redacted Subject: Highlight Date: 12/16/2017 12:00:50 PM -05'00'

Author: redacted Subject: Sticky Note Date: 12/16/2017 12:02:43 PM -05'00'
Add space.

Author: redacted Subject: Highlight Date: 12/16/2017 12:02:16 PM -05'00'

Reviewers' comments:

Reviewer #1 (Remarks to the Author):

Harney et al review

In this manuscript Harney et al. present new genome-wide SNP data from 22 individuals from the Levant (Israel) dating to the Chalcolithic. They compare these samples primarily to other Levantine, Anatolian and Iranian genomes from the Neolithic through to the Bronze age, with the main aim of exploring whether at least two archaeological shifts that occur between these two points correlate with shifts in genetic ancestry, which may implicate population turnover and migration from outside the region. The authors conclude that Chalcolithic Levant populations as represented by this one specific site show genetic evidence of being a mixture of early Neolithic farmers from the region, Iranian Chalcolithic, and somewhat surprisingly, Anatolian Neolithic farmers, thus supporting the conclusion of migration. The authors also suggest only continuity with some Bronze Age populations from the same area.

This data produced in this paper is unique with regard to the scale of samples from a single site, especially one so old. The authors use a series of established but powerful methods that they have developed themselves to perform the demographic analysis. I imagine both the results and data will be of interest to those working on ancient DNA, population geneticists, archaeologists and anthropologists, given how important this part of the world is to understanding human prehistory out of Africa. In general the results seem robust but I do have some major issues that need addressing.

I note that I am not qualified to evaluate the archaeological claims.

Major Issues:

1. The primary evidence of migration into the area and population replacement suggested by the authors between the Neolithic and Chalcolithic (and then the Bronze age) that would explain changes in archaeological complexity hinges on new ancestry components entering the Chalcolithic Levant, and thus modeling them as a mixture of Neolithic Levant and other sources. However, I worry to what extent treating the Neolithic Levant samples as a single population is influencing some of these results. Is this really justified? My understanding is that the Neolithic Levant samples come from two locations and from multiple time periods during the Neolithic. There is a clear overlap for some samples between the Neolithic and Chalcolithic Levant in the PCA. In addition, in the Admixture analysis there are some Neolithic Levant individuals that appear to match Chalcolithic individuals, with the same ratio of blue and green component, while some Neolithic samples have zero green component that is so prominent in Neolithic Iranian populations. There are even two individuals that have what looks like a component (red) found only in European and Anatolian Neolithic samples. Could it be that

Chalcolithic Levant (and perhaps only individuals from this particular site, rather than the region in general) simply emerged from one part of a structured Neolithic Levant population, and that no major migration into the region is needed to explain the results (particularly from the Iranian side)? I'd would like to see the authors test to what extent population structure is influencing the results, either redoing the various qpAdm analysis with individual Neolithic Levant individuals or grouping by sampling location and time period. Perhaps the original results will still emerge, but it should be tested.

We thank the referee for this suggestion, and we have now looked into these issues carefully. There is no evidence that the Levant Neolithic samples we analyzed are substructured in a way that overlaps the genetic variation in the Levant Chalcolithic samples, as we describe below.

The Neolithic Levantine samples come from two sites: Motza, Israel (n=1) and 'Ain Ghazal, Jordan (n=12). The sample from Motza is categorized as PPNB, and those from 'Ain Ghazal are categorized as both PPNB (n=10) and PPNC (n=2). Despite the geographic and temporal heterogeneity of these samples, formal symmetry testing found that they were consistent with being a homogeneous group relative to other Near Eastern populations to the limits of the resolution of the study in which they were reported (Lazaridis et al. Nature 2016). The genetic homogeneity is consistent with their material cultural similarity.

Nevertheless, we agree with the referee that it is valuable to explore whether substructure in the Levantine population could be influencing the results, and we therefore repeated the analyses, separating the samples based on geography and time into three subsets:

- (i) Motza (n=1)**
- (ii) 'Ain Ghazal PPNB (n=10)**
- (iii) 'Ain Ghazal PPNC (n=2)**

We also separated the samples based on their ADMIXTURE results into two subsets:

- (A) Levant Neolithic samples with <5% assignment to the green admixture component (n=10)**
- (B) Levant Neolithic samples with >5% assignment to the green admixture component (n=3)**

First, we calculated symmetry statistics of the form (Levant_N_subset1, Levant_N_subset2; Test, chimp) for all combinations of subsets, for all ancient and present-day populations, *Test*. No *Test* contributed a Z-score $\geq |3|$, suggesting that it is appropriate to treat these subsets as a single homogenous population, as they are symmetrically related to all other *Test* populations. We report this in the revised text.

To be certain that undetected structure in the data is not biasing the results, we also repeated the *f*-statistic and *qpAdm* analyses using these subsets. While the significance level of each statistic was impacted—due to the reduction in statistical power caused by the smaller sample sizes—we observed no qualitative differences in the directionality of the affinities measured by each statistic between subsets.

Based on these results, we conclude that combining the 13 Levant Neolithic samples is valid, and that dividing the samples into subgroups would not qualitatively change our findings.

2. The qpAdmix (and qWave) sections that form the bulk of the results after the PCA and Admixture analysis are very difficult to read. It is hard work to follow the various tests being conducted and how they fit in to the larger picture. I know the authors developed these methods and thus can construct elaborate designs using various combinations of samples, but for people outside the field in particular, this section will be very hard to work through. Some figures showing the various models being tested would be useful, even if they get put in the supplementary material.

We thank the referee for encouraging us to clarify this. We have heavily rewritten the parts of the manuscript that discusses the qpAdm analyses to be more accessible, and we have added a section to the supplementary information breaking the analysis down and providing visuals to add further clarity. We think that the manuscript is now much more readable

3. One of the most unique aspects of this study is the generation of genomic data from a large sample from a single site, something the authors themselves highlight. However, very little is actually done with this, other than to say they are “homogenous”? This is disappointing. There does not seem to be much they have done with 21 samples that they could not have done with 1 or two samples. While I know this is only SNP-capture data, are there any inferences that could be made about population diversity or levels of inbreeding. It seems a waste. Also, what about population frequencies of phenotypic markers that I believe are within the captured SNP set based on a previous manuscript by this group (Mathieson et al. 2015)?

We have added a section on the frequencies of SNPs of potential biological interest. The results are included in the newly added Supplementary Table 9, comparing frequencies of Levant_ChL with other ancient populations (Natufian, Levant_N, Levant_BA_South, Levant_BA_North, Iran_ChL, and Anatolia_ChL) and present-day populations from the 1000 Genomes Project. An explicit discussion of several polymorphisms that we feel are of greatest interest to readers is included in the text.

We also performed an additional analysis in order to estimate genetic diversity, computing relative heterozygosity in the Levant_ChL population compared with all other ancient Levantine populations. We observe increasing within-population genetic diversity over time, with samples from the Paleolithic and Bronze Ages exhibiting the lowest and highest levels of genetic diversity respectively, a pattern that is driven by population mixture.

Specific Minor Issues.

Line 47: “One hypothesis is that the Chalcolithic culture in the region was spread by immigrants from the north, based on similarities in decorative methods”

Please be more specific about “north”. There is a lot of land up there!

The term 'north' was used by the cited authors. It is clear from the material culture and archaeological sites they mentioned that they are generally referring to north Mesopotamia. We specify this in the revised text.

Line 58: “It has been estimated that the burial cave contained up to 600 individuals”
Given the dimensions of the cave given above, this seems a lot. Is there a picture or figure of the site we could see?

In the revised manuscript we have included a picture of the cave (Figure 1b). The cave was indeed small and the ossuaries were piled one on top of each other. This was also the case for the skeletal material found outside the ossuaries.

Line 60: “Direct radiocarbon dating suggests that the cave was in use throughout the Late Chalcolithic period (c. 4500–3950 BCE), functioning as a secondary burial cemetery...suggests that it functioned as a central cemetery for the settlements located in the western Upper Galilee”

This is confusing, is it a secondary burial cemetery or a central cemetery? Please clarify (perhaps it can be both, though I’m not immediately sure how”)

Secondary burial refers to the practice of reintering remains that have previously been buried and were later exhumed and reprocessed, in this case placed in large containers called ossuaries. The use of the term “secondary burial site” thus does not mean that the site is of secondary importance, but instead means almost the opposite (a place where the remains were centralized). To avoid confusion, in the revised paper we only mention that the cave functioned as a central burial location and added references for those who are interested in more details.

Line 103: “The dataset is of extraordinarily quality”

Though I very much appreciate the value of the data, this is a bit hyperbolic. It is only “extraordinary” if placed into some context with samples from similar time periods. Please either rephrase or contextualize.

We have rephrased this as “The dataset is of unprecedented quality given the difficult preservation conditions in the warm Near East”

Line 118: “The Levant_ChL cluster falls in between the clusters corresponding to the Levant Neolithic (Levant_N) and Levant Bronze Age (including samples from ‘Ain Ghazal, Jordan (Levant_BA_South) and Sidon, Lebanon (Levant_BA_North)).”

From my examination of the PCA, it looks like the Levant ChL significantly overlap with Levant_N rather than being in between. Also, are all the other ancient (and even modern) samples from Europe necessary for this PCA? It would be much easier to understand the

pattern if we could zoom into the region and samples of interest. Everything else is a bit of a distraction.

We have updated the text to indicate that the Levant_ChL and Levant_N clusters overlap in the PCA (which we believe is likely to be statistical noise due to limited data from particular samples as there is no statistical evidence of heterogeneity within these clusters). Additionally, we have made some stylistic changes to the PCA plot so that the Levant_ChL samples stand out more clearly.

We feel that showing both Europeans and Near Easterners in the PCA plot is important. because the European samples were used to make the plot. In addition, the full plot showing both Europeans and Near Easterners is the same as that in relevant previously studies (e.g. Lazaridis et al 2016 and Haber et al, 2017), making it easier for readers to compare the different articles.

Line 128: “Figure 1d shows the ADMIXTURE results for the ancient individuals assuming $K=11$ clusters (we selected this number because it maximizes ancestry components that are correlated to ancient populations from the Levant, from Iran, and European hunter-gatherers)”
Could the relevant individuals be grouped together a little bit better, in particular Iranian Neolithic samples being close to the Levant samples? Also, the point must be made in the main text (see above) that some Levant Neolithic samples appear to show the same pattern as the Levant ChL samples (perhaps corresponding with their overlap in the PCA).

We have rearranged the order of the populations in the ADMIXTURE plot so that the populations of greatest interest are placed close together, while still generally grouping populations with similar ancestry profiles close together. Additionally, we have added a discussion of the statistical indistinguishability of some Levant_ChL and Levant_N samples, and highlighted that despite this similarity in ADMIXTURE results, this does not represent statistically significant structure within the Levant_N samples to the limits of our resolution.

Line 144: “Figure 2a shows that Levant_N shares more alleles with the Levant_ChL than it does with any other population, as the statistic is always positive. However, Levant_ChL is also not directly descended from Levant_N”

As described above, I assume the Levant N samples are grouped together and analyzed simultaneously in this analysis. How does variation with Levant N affect these analyses (what if for example each Levant N sample is analyzed individually)? What is the impact of population structure within Levant N on the model, methods and interpretation?).

As outlined above, we observe no impact of subsetting the Levant_N samples, other than a loss in statistical power due to reduction in sample size. It is therefore appropriate to pool.

Also, not all these comparisons appear to be significant, in particular for European early and Anatolian Neolithic.

We have rephrased any references to statistical significance to make it clear that not all statistics are considered significant (greater than three standard errors away from zero). Instead, we now simply observe that Levant_ChL is closest to Levant_N by this approach.

Line 167: “To search for models that fit the ancestry of the Levant_ChL, we chose as Left populations Levant_ChL along with various subsets of the 5 populations that are geographically and temporally closest to the Levant_ChL (Anatolia_N, Anatolia_ChL, Armenia_ChL, Iran_ChL, Iran_N, Levant_N), in conjunction with other ancient populations.”
What does the “in conjunction” refer to in the analysis framework described in the previous paragraph? The ancestral populations? A bit confusing.

We have rewritten these confusing sentences.

Line 220: “particularly other ancient Levantine populations (Supplementary Table 6)”
Is Supp Table 6 correct, it just has one row in it?

We agree that this one-line table was odd, so we have added to it results for all statistics of the form $f_4(\text{Levant_BA_North}, \text{Levant_BA_South}; \text{Test}, \text{Chimp})$ for all ancient and modern populations, *Test*, rather than just reporting the largest and smallest statistics.

Line 281: “suggesting that the emergence of the Chalcolithic material culture was associated with population migration and turnover....The presence of Iran_ChL-related ancestry in both populations – but not in the earlier Levant_N – suggests migration into the Levant of populations related to Iranian farmers, which must have occurred at least by the time of the Chalcolithic. The Anatolian_N component present in the Levant_ChL but not in the Levant_BA_South sample suggests that there was also a separate migration of Anatolian-related people into the region.”

See major issue about population structure on whether these inferences are supported.

As we do not find evidence for detectable population structure in the Levant_N dataset (discussed above) we did not alter this statement, although in our revised manuscript we explicitly state that there may exist unsampled population structure in the Neolithic Levant.

Line 386: “We estimated F_{ST} using smartpca for the 21 ancient Near Eastern populations made” The results of this are never mentioned.

We have added a discussion of F_{ST} to the revised text.

Reviewer #2 Comments to the Authors

Not being an expert on DNA research, I will limit my comments to the archaeological aspects of the paper.

The southern Levant Ghassulian culture of the Chalcolithic period (ca. 4,500-3900 BC cal.) features many unique and innovative cultural attributes, as the authors indicate. The copper industry, for one, especially the use of the "lost wax" technique, is probably the earliest advanced metallurgy known. Since its discovery in the late 1920s, the thorny questions of the Ghassulian origin and its termination are central to the research of this time span. Until now the attempts to address this question were based essentially on "subjective" comparisons of iconography and artifacts' shapes. The proposed paper offers a totally different "objective" answers on the basis of comparing ancient DNA extracted from human bones from the cave of Peki'in in the Upper Galilee, Israel, to earlier and later populations from the southern Levant, as well as to populations from Iran and Anatolia.

The proposed paper advocates that the long debate is now settled: the ancient DNA shows that the uniqueness of the Ghassulian is due to an intrusive cultural entity. The genetic evidence indicates complex movements and local populations turnovers. Thus, it is argued that two waves of migrations, from Iran and from Anatolia, triggered the inception of the Ghassulian. Concerning the end of this culture, the authors argue that the succeeding Bronze Age population shows different genetic attributes and therefore represents a discontinuity, well reflected in the material culture that differs from the previous period.

The idea that the Ghassulian Chalcolithic was an intrusive culture has already been suggested in the past on the basis of equivocal typological grounds. However, the intrusion model is now supported by ancient DNA, and this is a novelty. The results of this study are of interest to a number of research communities including archaeologists, ancient DNA geneticists, anthropologists and historians of technology.

I recommend to publish the paper, but I suggest to consider corrections and comments added to the text which I hereby attach. I've also found mistakes in the list of references concerning the archaeological aspect of the paper. I suggest a number of corrections in the attached file and recommend a careful re-checking of the list.

Comment: All the defining attributes listed below characterize a segment of the "Late Chalcolithic" period. There are parts in the north and the south of the southern Levant, in which none of these distinctive characteristics are found. The listed attributes characterize the Chalcolithic cultural entity most authorities refer to as "Ghassulian". I suggest that the terms Ghassulian be mentioned in the text, at least in this introductory section.

We appreciate the referee's attention to the issues of terminology. To our understanding, however, "Ghassulian" is not accepted by all archaeologists of the Galilee as a term for

referring to Late Chalcolithic material cultures of the region. In our revised manuscript, we therefore highlight the connection to the Ghassulian but do not definitively identify Peki'in with the Ghassulian, writing "The distinctive cultural characteristics of the Late Chalcolithic period in the Levant (which is often related to the Ghassulian culture, although this term is not applied in practice applied to the Galilee region, where this study is based) have few stylistic..."

Comment: To add "...and the use of the "lost wax" technique,"

We have made this change to the text.

Comment: Perrot in his 1955 paper (reference 12) discusses the origin of the "Beersheba culture", and suggests that it originated in Trans-Jordan (ibid pp. 185-188). Concerning the other Chalcolithic sites, he claims that there are sites "...of local origin..." and a group of sites which "We do not know its immediate origins." Thus, this reference does not support the claim for a northern origin and should be removed... The same applies to reference 13. A northern origin of the "Ghassoulien" is not mentioned... Kenyon (1979: p. 63) suggest that the Ghassulians "...must have come from the east or north-east..."

There are diverse opinions regarding the geographical origin of the Ghassulian culture, and the original list of references was included to express this diversity in opinions. On reflection though we agree with the referee's suggestion and have omitted the references which do not claim northern origin for the Ghassulian culture (# 12, 13 and 16).

Comment: The "burial jars" of Peki'in are ossuaries.

The distinction we were trying to make between an ossuary and a burial jar was to separate between containers that were created intentionally for burial and domestic containers whose secondary usage was for burial. We have revised this sentence to clarify this issue as follows: "... ossuaries and domestic jars repurposed as ossuaries"

Comment: A problematic statement. The Peki'in cultural assemblages had strong cultural affinities with the Ghassulian culture (off-sites cemeteries in ossuaries, similar iconography, the same pottery vessel types, basalt bowls, violin-shaped figurines, copper artifacts etc.). There are also affinities with the Golan Chalcolithic, pottery of which is present at the cave. If the Peki'in cultural affinities are enigmatic, how do they contribute to understanding the genetic origins of the what the authors label "Late Chalcolithic" (Ghassulian)?

This comment is well taken. We deleted the sentence from the text.

Comment: 3960?

We agree with the reviewer that dates in this period should not be specified so exactly. We therefore changed the date to 3900 BCE.

Comment: Contradicts the statements in lines 64-65, concerning the "enigmatic" cultural affinities of Peki'in (see my comment there).

As suggested by the reviewer, we removed this sentence.

Comment: Add: de Vaux 1970: 529-530 (reference 15).

We have added this reference suggested by the referee.

Reviewer #3 (Remarks to the Author):

This is a report of a successful genomic analysis of a population sample from an interesting archaeological context - the Levantine Chalcolithic. Substantial genome wide data from 22 individuals are obtained, and these allow new inference about the complexity of migration and admixture in the ancient Near East. Some positives about this work are:

- the densities of coverage of the targeted SNPs are unusually good for this region and time. This makes this a valuable contribution for further studies. Is it worth speculating on why? Was this a limestone context, free from water disturbance etc.

We have added a brief discussion about why we suspect the ancient DNA analysis of these samples was successful. We highlight both the skeletal sampling approach targeting the petrous portion of the temporal bone, and the favorable conditions inside the limestone burial cave.

- The wider archaeological context is well discussed and referenced.

- The implication of different ancestral components and therefore discontinuity with earlier and later horizons is novel and interesting, pointing to population flux

- The analysis seems sound and draws on existing established methods.

As such I judge that this work has the basis for publication in the journal.

Some comments:

Comment: The inference relies strongly upon *qpAdm* - is there merit in addressing this question using alternative approaches, eg *Treemix*?

***TreeMix* is fitting models using the same f-statistics as *qpAdm*. As we highlight in the revised text, the advantage of *qpAdm* is that it does not require specifying a complete model for the joint history of all analyzed populations. This makes it possible to make meaningful observations about the population history of the samples, without worrying that the results are biased due to incorrect assumptions made about the deeper relationships.**

We also attempted to analyze this dataset using two other methods apart from *qpAdm*—not only *TreeMix* but also *qpGraph*—which both specify more complete models. We found in practice that both tools were strongly impacted by assumptions about the underlying topology, which we could not confidently resolve and which in fact did not need to be resolved in order to make the arguments we make. Therefore, we relied mostly on *qpAdm*.

Comment: This unveils a new population sample from a region of strong interest, is there scope for asking if this population has made a specific contribution to later populations, eg in North Africa.

Lazaridis et al (Nature 2016) implemented a related analysis, and found that the Levant_N and Levant_BA_S populations could be modeled as plausible source of West Eurasian related ancestry in all but two and all but four East African populations, respectively. We repeated this analysis using the Levant_ChL population and found that all but three East African populations could be modeled as an admixture between Levant_ChL and the ancient African Mota sample. Therefore, it appears that while the Levant_ChL population is likely related to the population that contributed West Eurasian related ancestry in East African populations, it does not appear to be a better source of this ancestry than previously analyzed ancient Levantine populations. We have added a section to the supplement reporting this analysis.

Comment: The unusually strong recovery of genome-wide data should make possible the assessment of how variation maps across the genome. Are there any regions showing outlying diversity that may be signals of selection in this population?

We have added a discussion of the allele frequencies of sites of potential biological interest. We were not able to devise tests of natural selection based on this analysis, however, as this requires a model of how present-day populations descend from the ancient ones we are analyzing (as in Mathieson et al., Nature 2015) and we did not have such a model.

Comment: A small quibble, the reference given in the main text for the petrous bone's aDNA preservation qualities is Pinhasi et al. whereas in fact the primary reference is Gamba et al.

We have updated this reference.

REVIEWERS' COMMENTS:

Reviewer #1 (Remarks to the Author):

The new additions help the manuscript, in particular the new supplementary materials section on the qpAdm modeling, really help. I have just a few further minor comments, but otherwise this is acceptable for publication in my opinion.

I think Figure 3 should be Figure 2 in terms of text order (and vice-a-versa)

Line 105: "We replicate previous reports of dramatic declines of genetic differentiation over time in the Near East²⁴, observing a median pairwise F_{ST} of 0.023 between the Peqi'in samples (abbreviation: Levant_ChL) and other Near Eastern Neolithic and Chalcolithic populations."

I don't see how giving just the median F_{ST} helps supports the statement here without any context. It would be better if the authors better described the relevant row/column in the supplementary figure. I assume they are referring to the darker colors (i.e. lower F_{ST} s) for the Levant ChL column than the Levant ChL row? Would be good to highlight this in the figure also.

Line 325: "history of at least periods of population movements"
Missing a word

SI6 is not mentioned in the text, what is it's purpose.

In regards to the response to the question of testing for structure in the Neolithic Levant, unless I am missing something, I fail to actually see any of the new results in either the manuscript or supplementary information/tables

The closest thing I can see is this:

"and these individuals are not significantly genetically distinct from the other individuals included in Levant_N by formal testing"

I think the results of this testing should actually be provided in full in the manuscript/supplementary info, not just described for the reviewer.

Reviewer #3 (Remarks to the Author):

I am happy that my concerns have been addressed.

REVIEWERS' COMMENTS:

Reviewer #1 (Remarks to the Author):

The new additions help the manuscript, in particular the new supplementary materials section on the qpAdm modeling, really help. I have just a few further minor comments, but otherwise this is acceptable for publication in my opinion.

I think Figure 3 should be Figure 2 in terms of text order (and vice-a-versa)

The figure names have now been switched, as suggested.

Line 105: "We replicate previous reports of dramatic declines of genetic differentiation over time in the Near East²⁴, observing a median pairwise F_{ST} of 0.023 between the Peqi'in samples (abbreviation: Levant_ChL) and other Near Eastern Neolithic and Chalcolithic populations." I don't see how giving just the median F_{ST} helps supports the statement here without any context. It would be better if the authors better described the relevant row/column in the supplementary figure. I assume they are referring to the darker colors (i.e. lower F_{ST} s) for the Levant ChL column than the Levant ChL row? Would be good to highlight this in the figure also.

We agree that this was unclear and have updated the paragraph to add context, it now reads:

"We replicate previous reports of dramatic declines of genetic differentiation over time in West Eurasia²⁴, observing a median pairwise F_{ST} of 0.023 (range: 0.009-0.061) between the Peqi'in samples (abbreviation: Levant_ChL) and other West Eurasian Neolithic and Chalcolithic populations, relative to a median pairwise F_{ST} of 0.098 (range: 0.023-0.153) observed between populations in pre-Neolithic periods, 0.015 (range: 0.002-0.045) in the Bronze Age periods, and 0.011 (range 0-0.046) in present-day West Eurasian populations²⁴. Thus, the collapse to present-day levels of differentiation was largely complete by the Chalcolithic (Supplementary Figure 1). "

Additionally, we have added further detail to the Supplementary Figure 1 legend, which now reads:

"As previously reported, between population differentiation is generally greater among more ancient populations (top and left), while more recent populations (bottom and right) exhibit less between population differentiation. The Levant_ChL population exhibits the greatest affinity to populations closely related to those from which it descends (ie. Levant_N, Anatolia_N, Iran_ChL) and later Bronze Age Levantine populations (i.e. Levant_BA_Noah and Levant_BA_South)"

Line 325: "history of at least periods of population movements"
Missing a word

We have edited this line to read “history of at least a few periods of population movement”

SI6 is not mentioned in the text, what is its purpose.

We have added a reference to SI6 (now Supplementary Note 3) in the discussion, which was added to respond to comments by reviewer 3. It reads: “We additionally find the Levant_ChL population does not serve as a likely source of the Levantine-related ancestry observed in present-day East African populations (see Supplementary Note 3)²⁴.”

In regards to the response to the question of testing for structure in the Neolithic Levant, unless I am missing something, I fail to actually see any of the new results in either the manuscript or supplementary information/tables

The closest thing I can see is this:

"and these individuals are not significantly genetically distinct from the other individuals included in Levant_N by formal testing"

I think the results of this testing should actually be provided in full in the manuscript/supplementary info, not just described for the reviewer.

As these results are not novel (formal testing was previously done in Lazaridis et al, 2016) we do not feel that it is appropriate to repeat them in the main text. Nevertheless, we have added a section to the supplementary information detailing these results.

Reviewer #3 (Remarks to the Author):

I am happy that my concerns have been addressed.